# Massive Activations in Large Language Models

**Mingjie Sun**[1*] **Xinlei Chen**[2] **J. Zico Kolter**[1,3] **Zhuang Liu**[2]
[1]Carnegie Mellon University  [2]Meta AI Research  [3]Bosch Center for AI

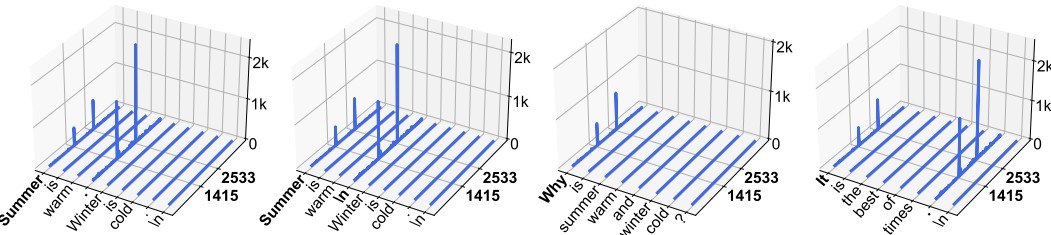

Figure 1: **Activation Magnitudes (z-axis) in LLaMA2-7B. x** and **y** axes are sequence and feature dimensions. For this specific model, we observe that activations with massive magnitudes appear in two fixed feature dimensions (1415, 2533), and two types of tokens—the starting token, and the first period (.) or newline token (\n).

## Abstract

We observe an empirical phenomenon in Large Language Models (LLMs)—very few activations exhibit significantly larger values than others (e.g., 100,000 times larger). We call them *massive activations*. First, we demonstrate the widespread existence of massive activations across various LLMs and characterize their locations. Second, we find their values largely stay constant regardless of the input, and they function as indispensable bias terms in LLMs. Third, these massive activations lead to the concentration of attention probabilities to their corresponding tokens, and further, implicit bias terms in the self-attention output. Last, we also study massive activations in Vision Transformers. Code is available at https://github.com/locuslab/massive-activations.

## 1 Introduction

Large Language Models (LLMs) (Brown et al., 2020; OpenAI, 2023) have demonstrated remarkable capabilities. The majority of existing studies conducted on these models are focused on their external behaviors, e.g., evaluating their performance on various tasks (Katz et al., 2023; Bubeck et al., 2023), developing prompts to elicit accurate responses (Wei et al., 2022; Yang et al., 2023). While these studies are encouraging and highlight the potential of these models, it is also important to gain insights into their internal mechanisms, especially as they are being increasingly integrated into many real-world applications. However, research on the internal workings of these models remains relatively limited.

In this work, we discover and study a surprising phenomenon in the internal representations of LLMs. Examining the hidden states in these models, we find that certain activations exhibit huge magnitudes, e.g., more than 4 orders of magnitude larger than the median, and could take on absolute values larger than 15,000 in LLaMA2-70B (Touvron et al., 2023), despite the presence of normalization layers. These activations are also extremely rare, often numbering fewer than 10 among tens of millions of total activations. Figure 1 illustrates this phenomenon in LLaMA2-7B. As these activations are so much larger in magnitudes compared to others, we name them *massive activations*. We demonstrate their presence in a wide range of LLMs, spanning different model sizes and families.

We explore where massive activations are located in LLMs. Regarding the depth dimension of LLMs, the appearance of massive activations is mostly abrupt: they emerge suddenly

---

[*]Correspondence to mingjies@cs.cmu.edu and zhuangl@meta.com.

after a single layer of computation, and diminish at the last few layers. Further, we find massive activations occur in a small number of feature dimensions that are input agnostic. Many of these activations are found within the starting word token and delimiter tokens. Additionally, we show that massive activations are not the same as outlier features (Dettmers et al., 2022), a previously known phenomenon in LLMs.

We show that massive activations act as fixed but crucial bias terms in LLMs. Here by bias terms, we mean certain internal states of the models that are independent from the inputs, analogous to the bias term $b$ in a linear layer $y = Wx + b$. First, we show that massive activations play a critical role in LLMs' capabilities. For instance, in LLaMA2-7B, setting merely four massive activations (out of millions of activations) to zero would result in catastrophic collapse in model performance. Further, setting them to their mean values does not hurt the model, suggesting their role is equivalent to simple constant biases. Our analysis reveals that after the initial layers, LLMs repurpose the tokens linked with massive activations to store these important biases.

Intriguingly, massive activations are closely connected with self-attention. In particular, we show massive activations cause attention to be attracted to the tokens associated with them. Our findings extend the observations from "attention sinks" (Xiao et al., 2023b)—we demonstrate that LLMs allocate excessive attention to more than just the first token, and provide an in-depth analysis on how such attention concentration patterns arise. Our analysis suggests that LLMs try to learn implicit bias components in self-attention via massive activations, during pretraining. We thus experiment with augmenting self-attention with extra key and value embeddings, explicitly designed as biases. Remarkably, we demonstrate that training with them eliminates the need for LLMs to learn massive activations.

Finally, we also observe massive activations in Vision Transformers (ViTs) (see Appendix A). They appear in many of the ViTs we have examined. In these ViTs, they tend to appear at fixed feature dimensions, but notably at varying patch tokens. Moreover, we find that these activations act similarly as fixed biases. Notably, we relate these findings to the recently proposed "register tokens" (Darcet et al., 2023). We show that they both learn values independent of input images. This offers an alternative interpretation for register tokens, where they were originally hypothesized to aggregate global image information.

## 2   Massive Activations

We study autoregressive Transformers, which are built by a stack of $L$ decoding layers. Each layer $\ell$ takes the previous hidden state $\mathbf{h}_{\ell-1} \in \mathbb{R}^{T \times d}$ as input and outputs a hidden state $h_\ell \in \mathbb{R}^{T \times d}$. $T$ is the number of tokens and $d$ is the number of features. Transformer layers use residual connections (He et al., 2016), and the computation can be formulated as:

$$h_\ell = h_{\ell-1} + \mathcal{F}_\ell(h_{\ell-1}) \tag{1}$$

where $\mathcal{F}_\ell$ is the residual transformation. Note that this includes both attention and MLP blocks. An *activation* denotes a specific scalar value in a hidden state. Unless otherwise specified, our study of activations is on the hidden state $h_\ell$, i.e., the output of residual summations, not any intermediate states inside $\mathcal{F}_\ell$.

**Existence in LLMs.** We start with an illustrative example on LLaMA2-7B. In Figure 1, we visualize the intermediate features $\mathbf{h}_\ell$ of interest. We feed this model with short sentences and visualize the activation magnitudes ($\mathbf{z}$-axis) of the hidden states at a middle layer. $\mathbf{x}$ and $\mathbf{y}$ axes are sequence and feature dimensions respectively. Each blue row corresponds to the feature embedding of one token. We observe up to four activations with significantly large magnitudes. The largest activation (about 2,000) is approximately 10,000 times larger than the median magnitude (about 0.2). The sheer scale of these activations makes them stand out from others. We thus refer to these special activations as *massive activations*.

Massive activations are not unique to this specific model LLaMA2-7B, but are widely observed in LLMs. In Figure 2 and Figure 3, we demonstrate the existence of massive activations in both LLaMA2-13B and Mixtral-8x7B (Jiang et al., 2024). Notably for Mixtral-8x7B, the largest activation magnitude can reach an absolute value of 7,000, around 4 orders of magnitude larger than the median feature magnitude (around 0.3). We refer the reader to Appendix B for results on more pretrained and fine-tuned LLMs.

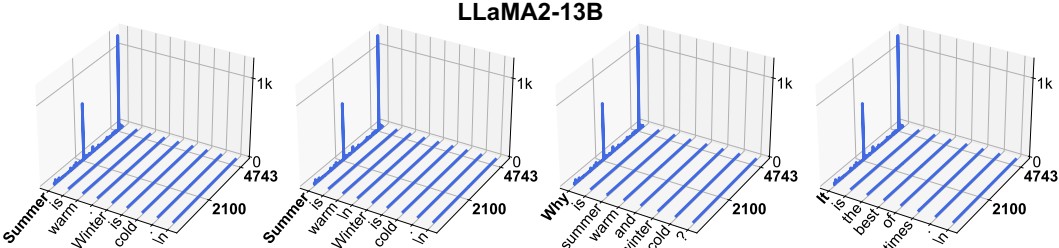

Figure 2: **Massive activations in LLaMA2-13B.** In this model, they appear in two fixed feature dimensions (2100, 4743), and are limited to the starting token.

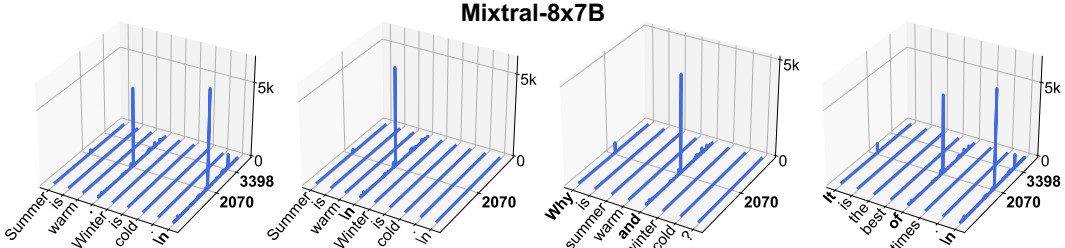

Figure 3: **Massive activations in Mixtral-8x7B.** In this model, they lie in two feature dimensions (2070, 3398), and are found within the starting token, delimiter tokens and certain word tokens ("and" and "of").

**Properties.** We summarize two main properties of massive activations. The most notable property is that these activations possess massive values and their magnitudes are significantly larger than other activations, often several orders of magnitude larger than the median value. Another property is that they are exceptionally few in number. For LLaMA2-7B in Figure 1, there are approximately 40,000 total activations in each presented hidden state but at most four massive activations can be identified.

Quantitatively, we present the values of the top activation magnitudes in Table 1. We also provide a loose but broad definition: an activation qualifies as a massive activation if its magnitude surpasses 100 and is at least or around 1,000 times larger than the median magnitude of its hidden state. We find this criterion to effectively identify these activations of interest across various LLMs, which are emphasized in bold in Table 1.

| Model | Top 1 | Top 2 | Top 3 | Top 4 | Top 5 | Top 10 | Top 100 | Top 1% | Top 10% | median |
|---|---|---|---|---|---|---|---|---|---|---|
| LLaMA2-7B | **2622.0** | **1547.0** | **802.0** | **477.3** | 156.9 | 45.7 | 10.6 | 1.1 | 0.6 | 0.2 |
| LLaMA2-13B | **1264.0** | **781.0** | 51.0 | 50.5 | 47.1 | 43.5 | 16.6 | 1.9 | 1.1 | 0.4 |
| Mixtral-8x7B | **7100.0** | **5296.0** | **1014.5** | **467.8** | **302.8** | 182.8 | 90.8 | 3.0 | 1.0 | 0.3 |

Table 1: Five largest, top 1% and 10%, and the median *activation magnitudes* at a hidden state of three LLMs. Massive activations in these LLMs are highlighted in bold.

Next, we identify the locations of massive activations within LLMs. For a comprehensive analysis, rather than using short sentences as inputs, we collect 100 sequences (each with 4,096 tokens) from RedPajama (Together Computer, 2023). We run LLMs on these 100 sequences and collect the hidden states from each layer.

## 2.1 Which Layers?

We determine the layers whose output hidden states exhibit massive activations. In Figure 4, we visualize the three largest activation magnitudes and the median of the hidden state output of each layer, with results averaged over 100 sequences. We examine three models: LLaMA2-7B, 13B and Phi-2 (Javaheripi et al., 2023) (see Appendix B.4 for more LLMs). In all cases, each of the top three activations comes from the same position in the hidden state across most of the middle layers. Generally, we observe the following:

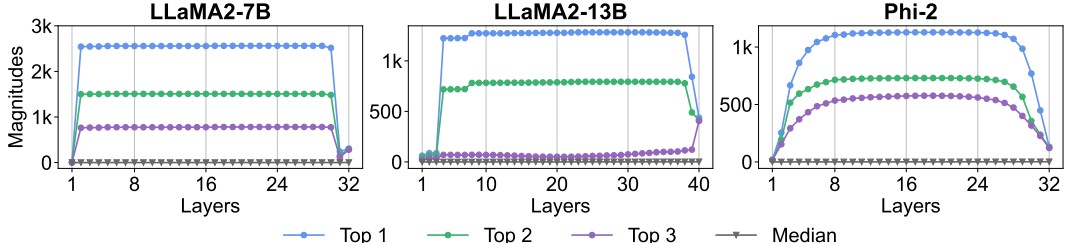

Figure 4: Three largest and the median activation magnitudes at each layer in LLMs.

*Massive activations exist and remain as largely constant values throughout most of the intermediate layers. They emerge in the initial layers and start to diminish in the last few layers.*

In LLaMA2-7B, massive activations first appear in layer 2 and remain nearly constant values until layer 30. Intriguingly, for LLaMA2-7B and 13B, massive activations emerge very rapidly from one layer of computation, e.g., layer 2 and layer 4 respectively. This means that they do not emerge as a result of gradual accumulation through many layers, and are caused by a rather different mechanism.

## 2.2 Which Feature and Sequence Dimensions?

We determine the locations of massive activations within hidden states, i.e., their feature and sequence dimensions. Since we have shown that their values largely stay constant in middle layers, we take on any such layer for this analysis.

**LLaMA2-7B.** In this model, massive activations are identified in two feature dimensions (1415 and 2533). Regarding sequence dimensions, we find that *massive activations appear at: 1. the starting word token, 2. the token representing the first period (.) or newline token (\n) in the sequence.* Figure 1 illustrates these findings for LLaMA2-7B. This is also consistent on long sequences. In cases where the input contains a "." or "\n" token, four massive activations are observed. For the less common scenario where neither "." nor "\n" is present, we can see two massive activations, both of which are associated with the initial token.

**LLaMA2-13B.** We find that massive activations in this model consistently appear in two feature dimensions, 2100 and 4743. *These activations are exclusively located within the starting token of the sequence, regardless of its semantics.* Figure 2 illustrates these behaviors within LLaMA2-13B. For *any* given input sequence, only two massive activations are present, corresponding to features 2100 and 4743 of the first word token.

**Mixtral-8x7B.** For this particular model, massive activations lie in two feature dimensions, i.e., 2070 and 3398. For sequence dimensions, we find that *they are associated with the starting token, delimiter tokens and also certain word tokens, e.g., token "and" and token "of".* Figure 3 showcases these patterns within Mixtral-8x7B. These word tokens tend to be conjunctions and prepositions, representing relatively few semantics of the input. Generally, for inputs of 4096 tokens in length, these tokens are predominantly located in the early part of sequence.

**Summary.** We summarize our findings of massive activations beyond the three models discussed above. We also put other models into categories based on empirical observations.

- For feature dimensions, they are consistently present in very few fixed dimensions.
- For sequence dimensions, we classify LLMs into three categories:
  a) Starting token only. Models include LLaMA2-13B, MPT and GPT-2.
  b) Starting token and the first "strong" delimiter token (i.e., "." or "\n"). Models include LLaMA2-7B and LLaMA2-7B-Chat.
  c) Starting token, delimiter tokens (such as ".", "\n", "'" or ","), and certain word tokens with weak semantics (such as "and", "from", "of" or "2"[1]). Models include LLaMA2-70B, Mistral-7B, Mixtral-8x7B, Falcon-40B and Phi-2.

---

[1]Such numeric tokens exhibit massive activations only in certain contexts, e.g., dates and years. Refer to Figure 17 for an illustration on LLaMA2-70B.

### 2.3 Difference from Outlier Features

With an understanding of the nature and locations of massive activations, we now discuss the differences between them and outlier features, a seemingly similar phenomenon in LLMs. Dettmers et al. (2022) have discovered outlier features in LLMs, characterized by large magnitudes. *Conceptually, a massive activation is a scalar value, determined jointly by the sequence and feature dimensions*; in contrast, *an outlier feature is a vector, corresponding to activations at all tokens*. Further, massive activations are present at extremely few tokens, while outlier features expect most activations in them to be large.

*In practice, we find that massive activations do not overlap with outlier feature dimensions*. We identify outlier features in LLaMA2-7B and 13B using the definition in Dettmers et al. (2022): a feature is deemed as an outlier feature if activation magnitudes exceed 6.0 at more than 25% of layers and 6% of tokens, on more than 90 out of 100 sequences. We discover 10 and 25 outlier features in these two models respectively. However, none of them correspond to the feature dimensions of massive activations.

## 3 Massive Activations Act as Biases in LLMs

While we have demonstrated the existence of massive activations and identified their locations, their functional role within LLMs is not yet clear. Are they important for internal computation? Or are they simply redundant activations with no effect? This section will delve deeper into LLMs to answer these questions. Different from the previous passive observations, we take a more proactive approach by inspecting how modifying massive activations affects the external behavior of LLMs.

| Model | Top 1 | Top 2 | Top 1% | Top 10% | Median |
|---|---|---|---|---|---|
| LLaMA2-7B | $2556.8 \pm 141.0$ | $-1507.0 \pm 83.0$ | $-0.14 \pm 0.6$ | $0.0 \pm 0.5$ | $0.2 \pm 0.3$ |
| LLaMA2-13B | $-1277.5 \pm 14.6$ | $-787.8 \pm 8.0$ | $0.9 \pm 0.7$ | $-0.3 \pm 0.8$ | $-0.3 \pm 0.6$ |

Table 2: The mean and variance of activation values at various positions, corresponding to the 2 largest, top 1% and 10%, and the median magnitudes within the hidden state. Variations in massive activations are significantly lower in comparison to other activations.

We first measure the variances of massive activations across input sequences. Besides massive activations, we choose three other positions based on their average magnitudes, corresponding to the top 1%/10%, and the median within the hidden state. In Table 2, we show the mean and standard deviation of the activation values at these positions across 100 sequences, for LLaMA2-7B and 13B. We find that the variances of massive activations are considerably smaller relative to their mean values when compared to other activations.

We then modify the inference of LLMs by intervening massive activations at one layer—for a hidden state exhibiting massive activations, we manually set these activations to chosen fixed values. Then the altered hidden state is fed into the next layer, and the computation afterwards continues as normal. We modify massive activations in LLaMA2-7B and 13B. We evaluate the perplexity on WikiText, C4 and PG-19 and the mean zero-shot accuracy on BoolQ, PIQA, WinoGrande, Arc-Easy and Arc-Challenge. For each model, we perform the intervention once on the hidden state where massive activations first appear. This corresponds to layer 2 and layer 4 in LLaMA2-7B and 13B respectively.

**Setting massive activations to zero.** We evaluate the performance of LLMs without massive activations. We set their values to zero in the hidden state when they first appear, i.e., removing massive activations from intervened LLMs. The results (denoted by *Set to zero*) are

| | LLaMA2-7B | | | | LLaMA2-13B | | | |
|---|---|---|---|---|---|---|---|---|
| Intervention | WikiText | C4 | PG-19 | Mean Zero-Shot | WikiText | C4 | PG-19 | Mean Zero-Shot |
| Original | 5.47 | 7.85 | 8.57 | 68.95% | 4.88 | 7.22 | 7.16 | 71.94% |
| *Set to zero* | inf | inf | inf | 36.75% | 5729 | 5526 | 4759 | 37.50% |
| *Set to mean* | 5.47 | 7.86 | 8.59 | 68.94% | 4.88 | 7.22 | 7.16 | 71.92% |

Table 3: Intervention analysis of massive activations in LLaMA2-7B and 13B. We set massive activations to fixed values and evaluate the perplexity ($\downarrow$) and mean zero-shot accuracy ($\uparrow$).

shown in Table 3. Intriguingly, there is a significant degradation in model performance, e.g., exploding perplexity numbers. For comparative analysis, an equal number of activations— those with average magnitudes close to the median magnitude—are similarly set to zero. We find this leads to no performance drop. These results highlight the crucial role that massive activations play in the internal computation of LLMs.

**Setting massive activations to mean values.** We remove the small variances in the values of massive activations. Specifically, we adjust the values of massive activations to their empirical mean values. The means are computed on 100 sequences from RedPajama. The results of this intervention (denoted by *Set to mean*) are shown in Table 3. We find that there are negligible changes in perplexity and zero-shot accuracy. This shows that their values are constants and input agnostic, i.e., functioning similarly to bias terms.

To summarize our findings:

*Massive activations act as fixed but important biases in LLMs.*

**Why these layers and tokens?** The fact that these activations act as biases may explain why LLMs store them at certain layers and tokens:

- The tendency of these activations to appear at the starting token could be attributed to the fact that every autoregressive training instance contains an initial token. Since LLMs are based on next word prediction, the starting token is the only token used in all forward passes within a sequence.

- The existence of these activations in delimiter tokens might be due to the relatively low semantic value of these tokens, rendering them a low-cost option for storing such biases. Conversely, tokens with rich semantics would risk significant loss of input information, if they are repurposed to store biases.

- The fact that massive activations emerge only after a few initial layers might be because LLMs require some initial layers to process the meaning of the tokens associated with massive activations. At these layers, their semantics may be transferred to other token positions via self-attention, and preserved moving forward.

## 4 Effects on Attention

In this section, we study the internal mechanism of massive activations in LLMs, particularly in relation to self-attention.

### 4.1 Attention is Concentrated on Massive Activations

We observe a stark contrast in attention patterns when comparing layers before and after the appearance of massive activations in LLMs. Figure 5 shows the attention logits (before softmax), averaged over all heads per layer in LLaMA2-7B. The input is a prompt from MMLU (Hendrycks et al., 2021): "*The following are multiple choice questions (with answers) about machine learning.\n\n ...*". Recall that in LLaMA2-7B, massive activations first appear in the output of layer 2 (see Figure 4). We find that in layer 3 and deeper layers (e.g., layer 31), attention is mostly concentrated on the two tokens associated with massive activations. Our observations are also consistent across various LLMs. Figure 6 demonstrates such attention concentration patterns in LLaMA2-13B and Phi-2, on the same input. See Appendix C.1 for results on more LLMs.

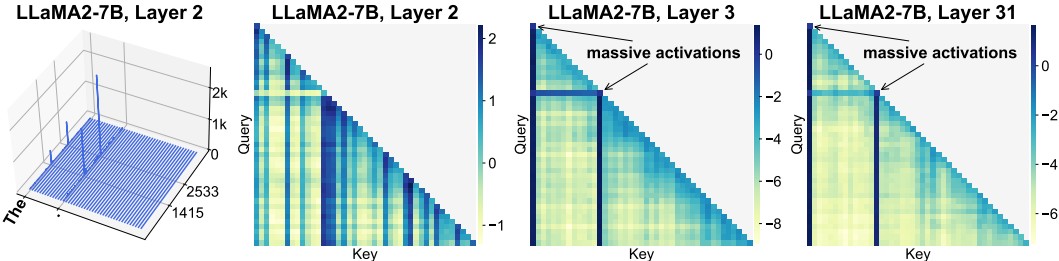

Figure 5: Attention patterns *before* and *after* massive activations appear in LLaMA2-7B. For each layer, we visualize average attention logits (unnormalized scores before softmax) over all heads, for an input sequence.

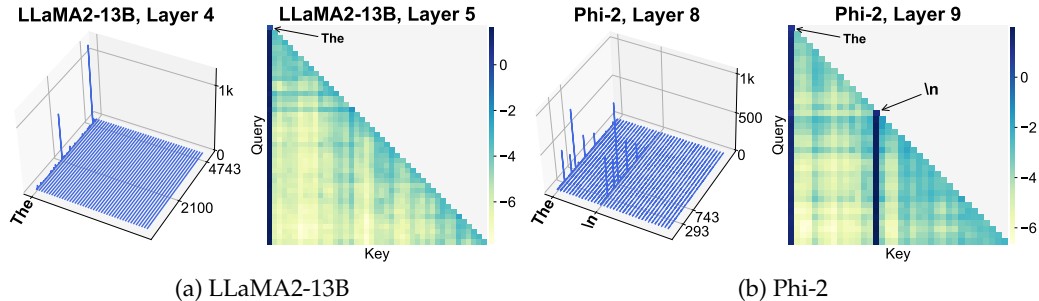

(a) LLaMA2-13B          (b) Phi-2

Figure 6: Attention patterns *after* massive activations emerge in LLaMA2-13B and Phi-2.

We notice that there is a consistent pattern across models on the distribution of attention logit values. In Figure 5 and Figure 6, many attention logits tend to be negative following massive activations. They are mostly computed by the inner product between query and key states of tokens without massive activations. However, when the key states belong to tokens associated with massive activations, the resulting attention logits are slightly positive. Thus in the attention softmax (computed along each row), these special attention logits will attract most of the attention probability.

Recently, Xiao et al. (2023b) showed that LLMs attend heavily to the starting token. Our findings on LLaMA2-13B in Figure 6a align with their results. Empirically, we find it is true for LLMs where massive activations are only found within the starting token. However, our results on LLaMA2-7B and Phi-2 indicate that LLMs also allocate substantial attention to other tokens and they are associated with massive activations. Furthermore, our results reveal a deeper cause for the emergence of these attention concentration patterns.

## 4.2 Massive Activations Impose Implicit Attention Biases

In this part, we delve into the computation within the attention block and demonstrate that LLMs use massive activations to enforce an implicit bias term in self-attention.

**Attention LayerNorm and QKV projections.** We study the impact of massive activations on the query, key and value states (Q/K/V) in self-attention. In LLMs, at each layer, input features are processed by layer normalization[2] (Ba et al., 2016) and then transformed into query, key and value states via linear projections, as illustrated in Figure 7a. This design choice is introduced in GPT-2 (Radford et al., 2019) and widely adopted in modern LLMs.

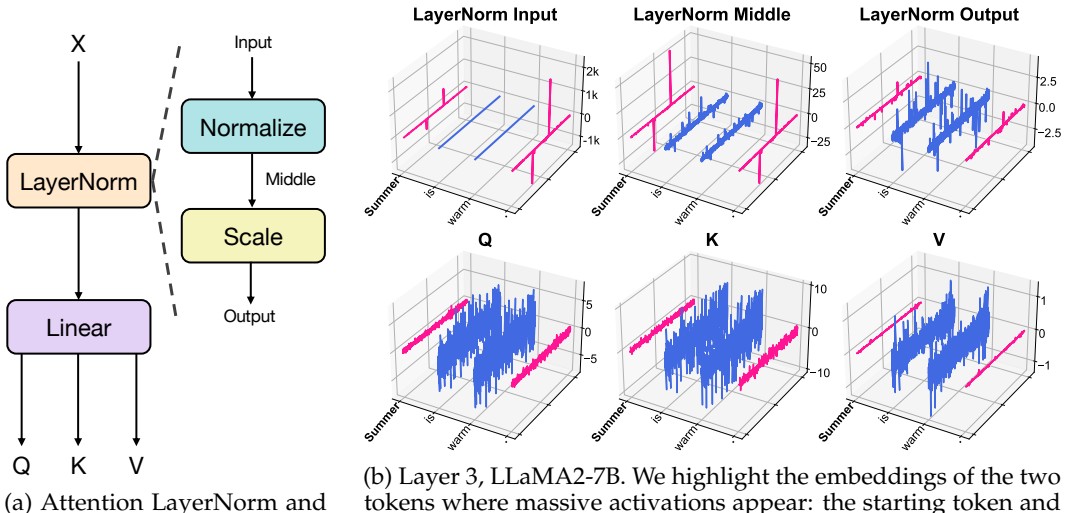

(a) Attention LayerNorm and QKV linear projections.

(b) Layer 3, LLaMA2-7B. We highlight the embeddings of the two tokens where massive activations appear: the starting token and the period token.

Figure 7: Activation trajectory from input hidden states to query, key and value states.

---

[2]LLaMA2 uses a variant of layer normalization: RMSNorm (Zhang & Sennrich, 2019).

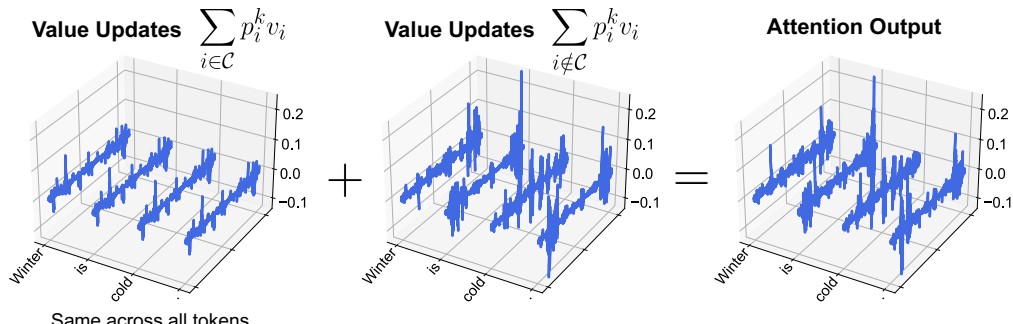

Figure 8: When computing the attention output, value updates computed from tokens associated with massive activations (the set $\mathcal{C}$) are essentially the same.

Figure 7b visualizes all hidden states computed in this schematic (LLaMA2-7B, layer 3). We find that at all stages, features of the two tokens associated with massive activations are drastically different from other tokens. Specifically, after the first "normalize" step, the embeddings of these two tokens appear as a sparse vector with two distinct non-zero elements. Notably, the subsequent QKV states exhibit considerably smaller variations within each embedding. We hypothesize that the attention LayerNorm may play a pivotal role in this process (see Appendix C.2 for further discussion).

**Attention output decomposition.** Given that attention is also concentrated on the tokens associated with massive activations (Section 4.1), we thus isolate these tokens and study their effects on the attention output (the layer of attention matrix multiplying value vectors). In Equation 2, we decompose the attention output at each token $k$ into two parts: value updates from the tokens $\mathcal{C}$ where attention is concentrated; and value updates aggregated from other tokens. Here $p_i^k$ is the attention distribution of query token $k$ to token $i$, and $v_i$ is the value state of token $i$.

$$\text{Attention}(Q, K, V)_k = \sum_{i \le k} p_i^k v_i = \sum_{i \in \mathcal{C}} p_i^k v_i + \sum_{i \notin \mathcal{C}} p_i^k v_i \tag{2}$$

Figure 8 visualizes the decomposed value updates and the attention output in LLaMA2-7B, with the input prompt "*Summer is warm. Winter is cold.*". In this case, the set $\mathcal{C}$ consists of token Summer and the first period token. We can see that the value updates from $\mathcal{C}$ are nearly identical across tokens, i.e., they serve as additive bias terms, although not explicitly imposed. Furthermore, we note that this pattern of value update is strikingly similar across various inputs. We refer the reader to Appendix C.3 for additional analysis. Overall, our results indicate that LLMs use massive activations to allocate substantial attention at certain tokens. These tokens are then utilized by LLMs to form a constant bias term when computing the attention output.

### 4.3 Explicit Attention Biases Eliminate Massive Activations

Given the strong need of LLMs to learn implicit attention biases during pretraining, we thus experiment with directly augmenting self-attention with additional bias terms. Intriguingly, we find that models with explicit attention biases do not exhibit massive activations.

**Formulation.** The idea is to model such attention biases explicitly, except not through repurposing existing tokens in the input sequence. Thus we introduce additional *learnable* parameters $\mathbf{k}', \mathbf{v}' \in \mathbb{R}^d$ for each head. Specifically, given input query, key and value matrices $Q, K, V \in \mathbb{R}^{T \times d}$, the augmented attention with explicit attention biases is computed as:

$$\text{Attention}(Q, K, V; \mathbf{k}', \mathbf{v}') = \text{softmax}\left(\frac{Q\left[K^T \ \mathbf{k}'\right]}{\sqrt{d}}\right)\begin{bmatrix} V \\ \mathbf{v}'^T \end{bmatrix} \tag{3}$$

where $\mathbf{k}'$ and $\mathbf{v}'$ are each concatenated with the key and value matrices K/V. The proposed attention can be used as a drop-in replacement of standard attention, without modifying other parts of Transformers, e.g., positional embeddings and MLP blocks.

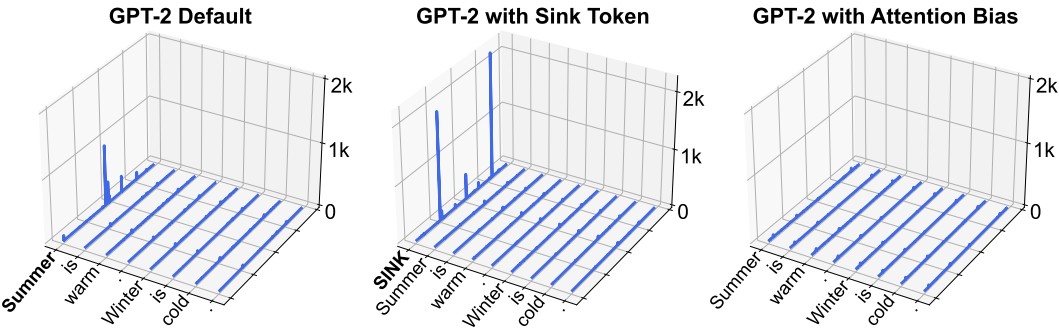

Figure 9: Massive activations disappear when training GPT-2 with explicit attention bias.

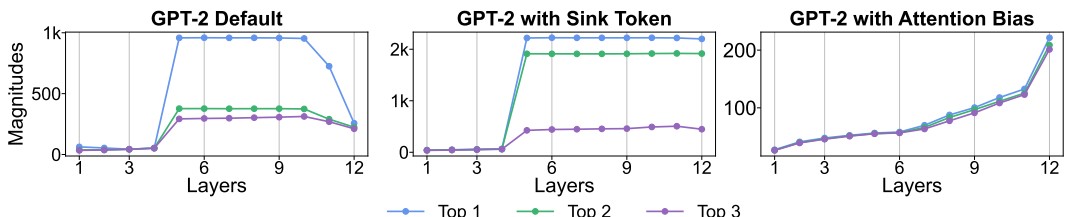

Figure 10: Three largest activation magnitudes at each layer for three GPT-2 models.

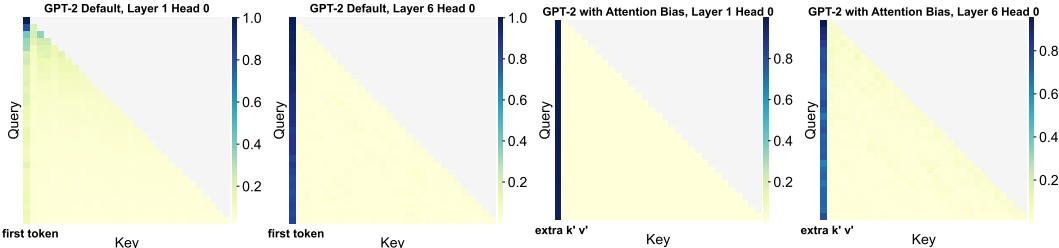

Figure 11: Attention distribution in default GPT-2 and GPT-2 with explicit attention bias.

**Results.** We train three GPT-2 models: the standard model, GPT-2 prepended with a sink token (Xiao et al., 2023b) and GPT-2 with explicit attention biases. See Appendix C.4 for training setups. We find that the three models have the same performance at convergence but differ significantly in the status of massive activations, as demonstrated in Figure 9. Notably, in GPT-2 with explicit attention biases, massive activations disappear, as compared to the default GPT-2 and one with a sink token.

Figure 10 shows the three largest activation magnitudes at the output of each layer in GPT-2. Notably, with explicit attention biases, top activation magnitudes in GPT-2 are increasing gradually as layers go deeper. In contrast, for both the default GPT-2 and GPT-2 with a sink token, massive activations emerge within one intermediate layer, i.e. layer 4. These results indicate that explicit attention biases negate the necessity for LLMs to develop massive activations during the pretraining phase.

In Figure 11, we visualize the attention distribution in both default GPT-2 and GPT-2 with explicit attention biases, where we plot the average attention probability over 50 sentences each with 30 tokens. First, we find that our observations on the relationship between massive activations and attention concentration hold for the default GPT-2 model. Second, for the GPT-2 model with explicit attention bias, most of the attention probability is assigned to the extra $\mathbf{k}'$ and $\mathbf{v}'$ vectors we inserted. Intriguingly, this also holds for initial layers as well (e.g., layer 1), suggesting the strong need for LLMs to form this attention concentration pattern during pretraining.

To summarize our findings in this section:

*Massive activations are connected to self-attention. LLMs use massive activations to concentrate substantial attention on very few tokens, injecting implicit bias terms in the attention computation. Further, massive activations can be eliminated by augmenting LLMs with explicit attention biases.*

## 5 Related Work

**Intriguing properties of autoregressive Transformers.** Timkey & Schijndel (2021) observed that in GPT-2's penultimate layer, there are feature dimensions containing activations with magnitudes up to 3,000, and these dimensions dominate standard measures of representation similarity. Heimersheim & Turner (2023) found that the feature norm of the initial token in GPT-2 grows much faster than other tokens. Kovaleva et al. (2021) and Zhao et al. (2023) discovered outlier weights in the LayerNorm of GPT-2 and LLaMA2-13B, and setting them to zero leads to catastrophic drop in model performance. Notably, the feature dimension of this weight in LLaMA2-13B (i.e., 2100) corresponds to that of a massive activation (Figure 2).

**Outlier features.** Various works in quantization (Dettmers et al., 2022; Zeng et al., 2022; Xiao et al., 2023a; Lin et al., 2023; Ahmadian et al., 2023) have studied the existence of outlier features in LLMs. Dettmers et al. (2022) showed that outlier features have large activation values in most of their sequence dimensions. While massive activations can be seemingly similar to outlier features, we discussed their fundamental differences in Section 2.3. We show that massive activations can not be attributed to the existence of outlier features.

**Attention concentration patterns.** Clark et al. (2019b), Kovaleva et al. (2019) and Bondarenko et al. (2021) discovered that attention in BERT (Devlin et al., 2018) tends to focus on the "separate" token [SEP]. Xiao et al. (2023b) showed that LLMs assign most of the attention to the starting word token. Darcet et al. (2023) revealed the existence of attention artifacts in ViTs. Robinson et al. (2023) found sparse activation patterns in ViTs that attract attention to certain tokens. Our work provides an in-depth analysis as to why these patterns emerge, specifically in relation to massive activations.

**Biases in self-attention.** There can be various notion of biases in the self-attention mechanism. First, simple additive bias terms can be used in linear layers for computing the query, key and value states (Namazifar et al., 2023). Many LLMs do not enable additive bias terms for linear layers during the pretraining phase (Chowdhery et al., 2022). Second, position biases can be inserted in self-attention to encode positional information of each token (Su et al., 2021; Press et al., 2021). There are also variants of biases with manually designed softmax operator (Miller, 2023; Bondarenko et al., 2023; Hu et al., 2024). Our work reveals that LLMs, even with standard self-attention, would impose implicit bias components in attention computation through massive activations.

## 6 Conclusion and Discussion

Autoregressive training of large Transformers has brought significant advances in natural language processing. This study reveals the widespread existence of *massive activations* in these Large Language Models (LLMs). The values of these activations are input agnostic but crucial for model performance, despite their extremely rare quantity. We establish a close connection between massive activations and the self-attention mechanism. We show that LLMs use them to implement an implicit form of biases for attention computation. We hope our results contribute to a deeper understanding of today's large-scale foundation models.

We discuss some practical implications and future directions of this work. First, the presence of activations with large magnitudes has been widely known as a major challenge in effectively quantizing LLMs (Dettmers et al., 2022; Xiao et al., 2023a). This paper identifies a new type of outlier activations in LLMs, and we hope our findings will be valuable to researchers working on LLM compression. Second, attention maps that allocate excessive attention probabilities to a few fixed tokens may be undesirable for mechanistic interpretability (Olsson et al., 2022). Our proposed attention formulation could make the resulting attention maps in LLMs more interpretable, and benefit downstream applications (Darcet et al., 2023). Finally, our investigation of the new attention formulation focused on its effects on massive activations, and our experiments were limited to a small GPT-2 model due to computational resource constraints. It would be interesting to see how our results generalize to models at larger scales, and how our attention formulation could affect the training stability (Wortsman et al., 2023) of modern LLMs.

## Acknowledgments

We thank Sachin Goyal, Jeremy Cohen, Timothée Darcet, Koustuv Sinha and Mike Rabbat for valuable discussions. Mingjie Sun was supported by funding from the Bosch Center for Artificial Intelligence.

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

## A   Vision Transformers

In this section, we study if Vision Transformers (ViTs) (Dosovitskiy et al., 2021) exhibit massive activations. We note that while ViTs and LLMs are both based on self-attention, ViTs employ global token mixing, which contrasts with the autoregressive nature of LLMs.

**Massive activations in ViTs.** We explore several model families based on ViTs: CLIP (Radford et al., 2021), MAE (He et al., 2021) and DINOv2 (Oquab et al., 2024). We examine the ViT-L models from these families. The activation magnitudes in the penultimate layer for an input image are illustrated in Figure 12. We find that massive activations exist in CLIP and DINOv2 ViT-L, where we highlight the corresponding sequence dimensions. In these two models, there are extremely few activations (fewer than four) with significantly larger magnitudes than others. In addition, these activations are located in specific feature dimensions and appear in *random* patch tokens. However, we do not observe massive activations in MAE ViT-L. In this model, a feature dimension (927) exhibits uniformly large values across all tokens.

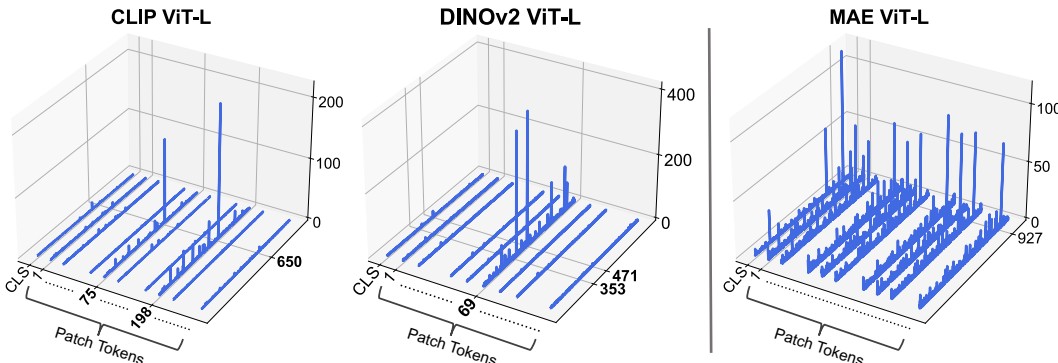

Figure 12: Massive activations are present in ViT-L from CLIP and DINOv2, but not MAE.

**Massive activations are biases in ViTs.** Figure 14 shows the three largest activation magnitudes and the median per layer in CLIP and DINOv2 ViT-L, averaged over 1k images. We find that massive activations are consistently present across images and their values remain largely the same around the mean values. It is worth noting that unlike LLMs, massive activations start to appear only in the later stages of ViTs.

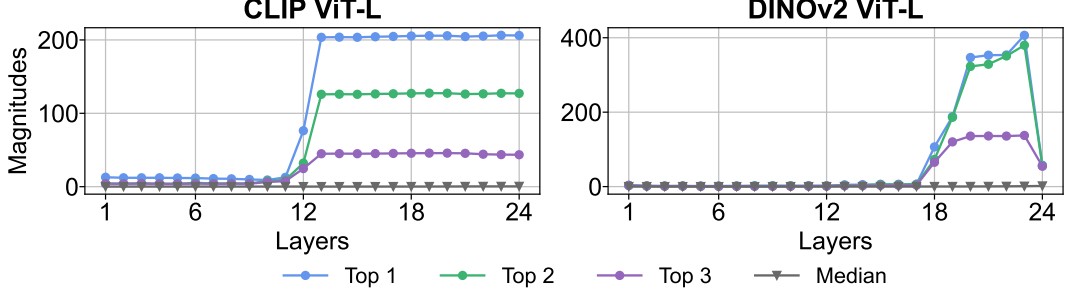

Figure 14: Three largest activation magnitudes and the median magnitude at each layer in CLIP and DINOv2 ViT-L.

|  CLIP ViT-L, layer 13  | |
| --- | --- |
| Intervention | ImageNet acc (%) |
| Original | 75.5 |
| *Set to zero* | 59.8 |
| *Set to mean* | 75.5 |

Table 4: Intervention analysis of massive activations in CLIP ViT-L.

Following Section 3, we perform intervention analysis on CLIP ViT-L. We modify the two largest massive activations to zero and mean values respectively. The intervention is

conducted on layer 13, where massive activations first appear within this model. Results are shown in Table 4, where we evaluate the zero-shot accuracy on ImageNet. We can see that setting massive activations to zero leads to significant drop in accuracy while setting to their means results in negligible accuracy drop. These results indicate that massive activations function as fixed but crucial biases in ViTs, aligned with our observations in Section 3.

**Registers are biases in ViTs.** Recently Darcet et al. (2023) propose to augment standard ViTs with additional learnable tokens, which they name as register tokens. They show that training ViTs with register tokens leads to smooth attention maps, and the resulting model family, namely DINOv2-reg, achieves superior downstream performance over DINOv2. Examining the largest ViT-G model in DINOv2-reg, we observe the existence of massive activations, as shown in Figure 15. However, different from standard ViTs, massive activations do not appear in patch tokens but exclusively within a fixed register token, i.e., register 3. This suggests that this model uses register 3 to store these activations. Figure 16 visualizes the attention distribution of the [CLS] token in the last layer. We find that most of the attention is allocated to register 3, echoing our findings in Section 4.1.

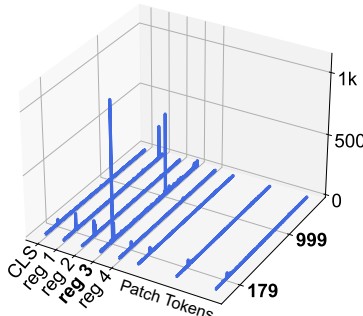

Figure 15: DINOv2-reg ViT-G.

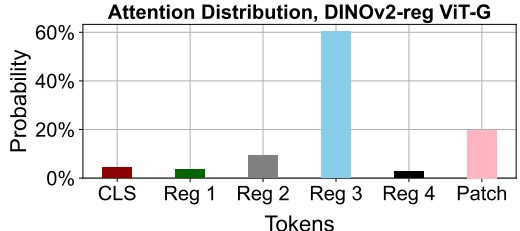

Figure 16: Attention probability of the [CLS] token, averaged over 1k ImageNet images.

| ImageNet acc (%) | DINOv2-reg with 4 registers | | | |
| --- | --- | --- | --- | --- |
| | ViT-S | ViT-B | ViT-L | ViT-G |
| Original | 81.9 | 84.8 | 86.3 | 87.0 |
| Fix-Reg-Mean | 81.7 | 85.0 | 86.2 | 87.0 |

Table 6: We fix *all* register features at *every* layer to their mean values and evaluate the intervened ViTs.

Further, we conduct intervention analysis to analyze the role of registers. We replace *all* register features at the output of *every* layer with their means, averaged over 10k ImageNet training images. This intervention removes the intended purpose of registers to aggregate global input information (Darcet et al., 2023). Table 6 shows the results. We find that ViTs with fixed register features achieve accuracy comparable to original models, suggesting that registers act as learned biases in ViTs. This leads to constant key and value states at register tokens, effectively introducing bias terms to self-attention (extra $\mathbf{k}'$ and $\mathbf{v}'$ in Equation 3). Thus a ViT with register tokens function equivalently to a standard ViT augmented with explicit attention biases.

In Masked Autoencoders (MAEs) (He et al., 2021), a dummy token is added to ViTs during pretraining. In one fine-tuning pipeline of MAEs, fine-tuning is done based on the average pooled features of all patch tokens. In these MAE models, this dummy token is equivalent to a register token. Here we maintain the register token features as constant across the output features of *all* layers in ViTs, which we denote as Fix-Reg-Mean. These fixed values are computed as the average register features over 10k ImageNet training images. Table 7 shows the results: setting register features to fixed values does not affect model performance.

| ImageNet acc | MAE with 1 register | | |
| --- | --- | --- | --- |
| | ViT-B | ViT-L | ViT-H |
| Original | 82.6 | 85.5 | 86.7 |
| Fix-Reg-Mean | 82.6 | 85.5 | 86.7 |

Table 7: Registers are biases in Masked Autoencoders (MAEs).

To summarize our findings:

*Massive activations exist in many but not all ViTs. Similar to those in LLMs, these activations act as constant biases. We also show the recently proposed register tokens have a similar function.*

## B  Additional Results on Massive Activations in LLMs

In this section, we supplement the main paper with additional results of massive activations in LLMs. This includes results on more pretrained LLMs (Appendix B.1) and fine-tuned LLMs (Appendix B.2), analysis of the BOS token  (Appendix B.3) and layer-level analysis (Appendix B.4).

### B.1  Pretrained LLMs

In Section 2, we have demonstrated massive activations in LLaMA2-7B, LLaMA2-13B and Mixtral-8x7B. In this section, we evaluate more pretrained LLMs which cover a wide range of model families. We illustrate massive activations in LLaMA2-70B, Phi-2, Mistral-7B (Jiang et al., 2023), MPT-7B (MosaicML, 2023) and Falcon-7B (Almazrouei et al., 2023). The results are presented in Figure 17, 18, 19, 20 and 21.

We make several observations. First, massive activations are consistently present in these models and they exhibit similar characteristics to those described in Section 2. Intriguingly, for LLaMA2-70B, we find that massive activations are found within tokens representing numerical values, e.g., token "0" and token "2", as depicted in Figure 17. However, they do not appear in all numerical tokens (see the *rightmost* example in Figure 17). Another interesting finding is that the feature dimension of massive activations in both Mistral-7B (Figure 19) and Mixtral-8x7B (Figure 3) is identical (i.e., 2070), implying that the latter model may have been fine-tuned from the former.

**LLaMA2-70B**

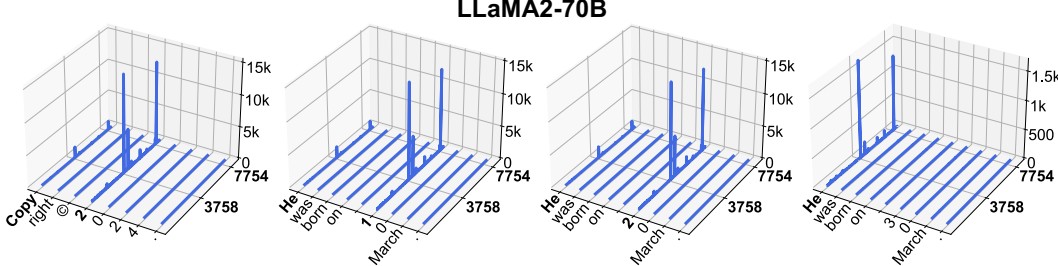

Figure 17: Massive activations in LLaMA2-70B.

**Phi-2**

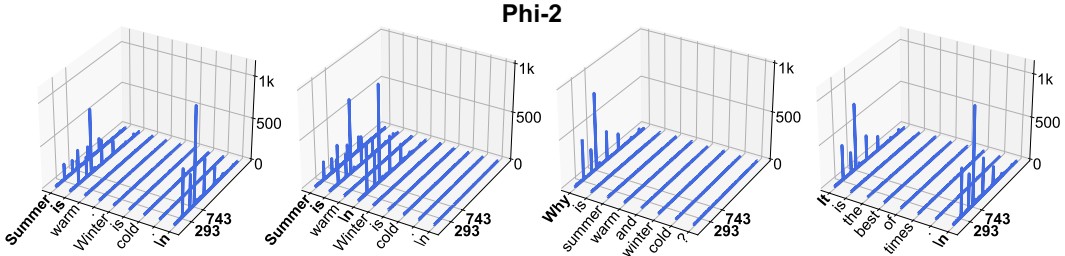

Figure 18: Massive activations in Phi-2.

**Mistral-7B**

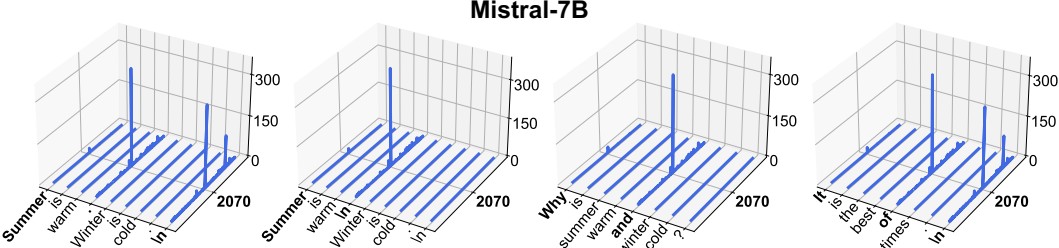

Figure 19: Massive activations in Mistral-7B.

### B.2  Fine-tuned LLMs

Our results so far are focused on pretrained LLMs. However, a significant application of LLMs lies in their use for chat purposes. Instruction fine-tuning (Ouyang et al., 2022) is essential for developing models capable of generating coherent responses to questions. In

**MPT-7B**

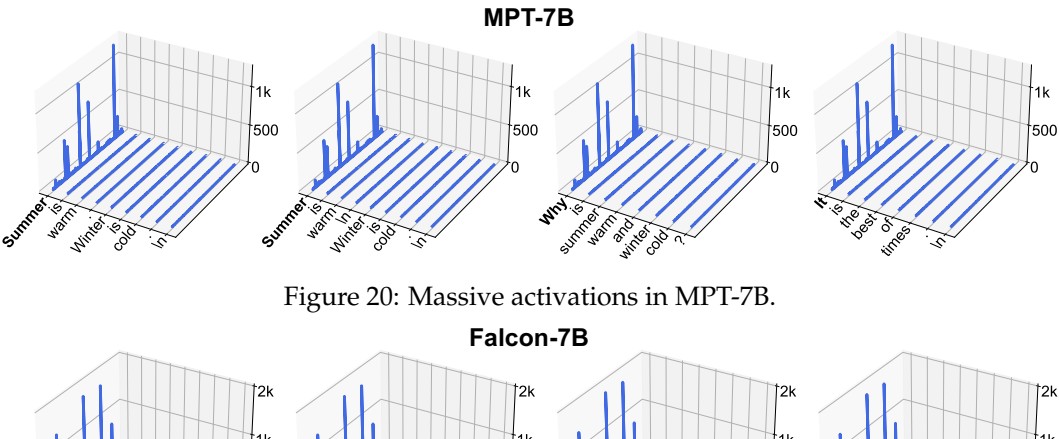

Figure 20: Massive activations in MPT-7B.

**Falcon-7B**

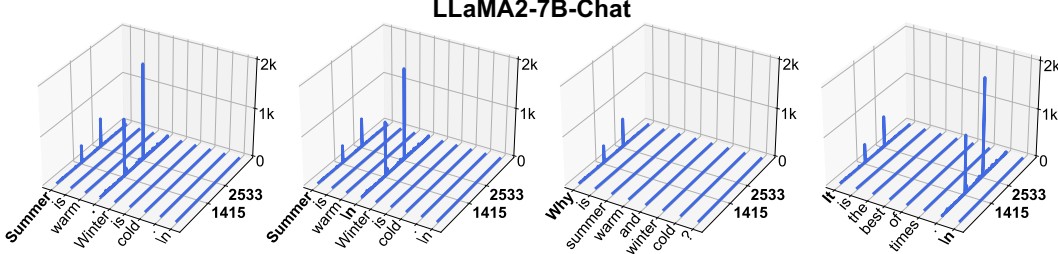

Figure 21: Massive activations in Falcon-7B.

this part, we demonstrate massive activations in these fine-tuned models. We evaluate fine-tuned models from models in LLaMA2 and Mistral. The results are shown in Figure 22, 23, 24 and 25.

We can see that massive activations persist after instruction fine-tuning. Moreover, the values and positions of massive activations remain largely the same as the original pretrained LLMs. For LLaMA2-7B, this can be seen by comparing Figure 22 and Figure 1. However, one exception is Mixtral-8x7B. We find that massive activations disappear from the newline token "\n" after fine-tuning, as shown by comparing Figure 25 and Figure 3. We leave the study on how instruction fine-tuning affects massive activations for future work.

**LLaMA2-7B-Chat**

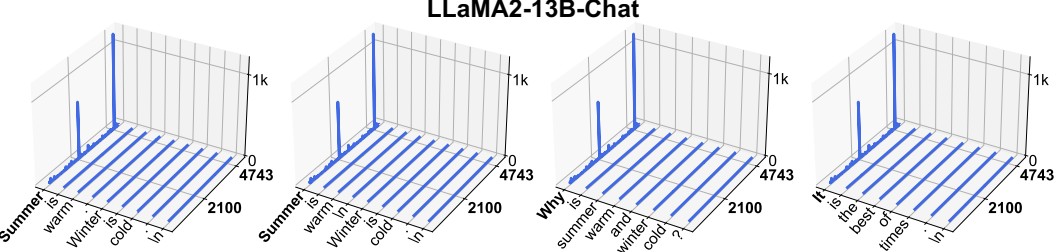

Figure 22: Massive activations in LLaMA2-7B-Chat.

**LLaMA2-13B-Chat**

Figure 23: Massive activations in LLaMA2-13B-Chat.

## B.3 BOS Token 

In some tokenizers, e.g., LLaMA2, the BOS token , also known as the beginning of sequence token, can be prepended to the input sequence. For the experiments presented in other parts of the paper, we turn off this option, where all sequences do not start with the BOS token.

**Mistral-7B-Instruct**

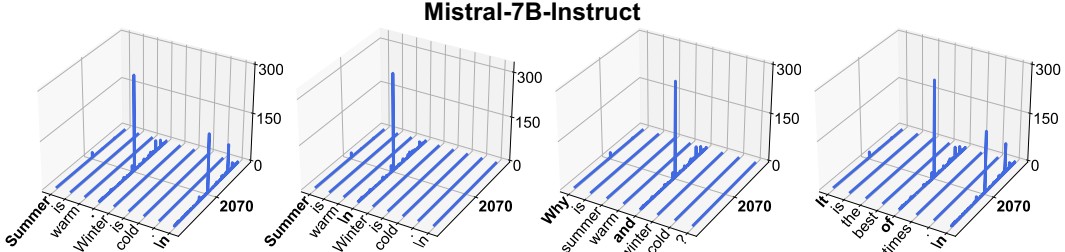

Figure 24: Massive activations in Mistral-7B-Instruct.

**Mixtral-8x7B-Instruct**

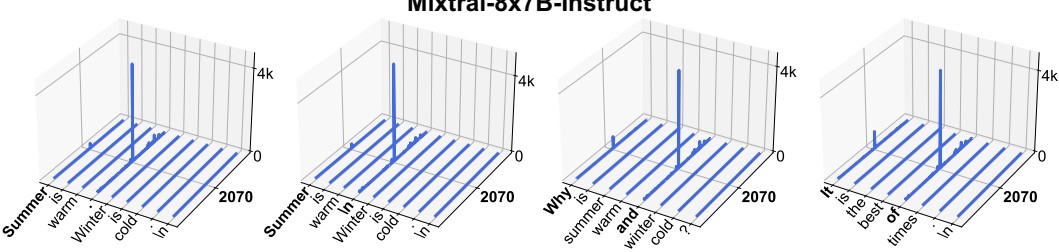

Figure 25: Massive activations in Mixtral-8x7B-Instruct.

In Figure 26, 27 and 28, we show massive activations in LLaMA2-7B, LLaMA2-13B and Mixtral-8x7B, with the same input sequences as in Section 2. We find that massive activations persist with a prepended BOS token. In LLaMA2-7B and LLaMA2-13B, the locations of massive activations, i.e., sequence and feature dimensions, are not altered. However, for Mixtral-8x7B, some massive activations shift to the BOS token . We leave the study on how the BOS token  affects the positions of massive activations for future work.

**LLaMA2-7B**

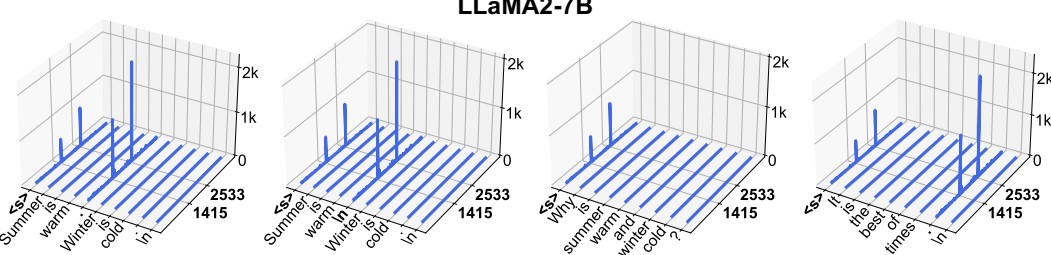

Figure 26: Massive activations in LLaMA2-7B when the input is prepended with a BOS token .

**LLaMA2-13B**

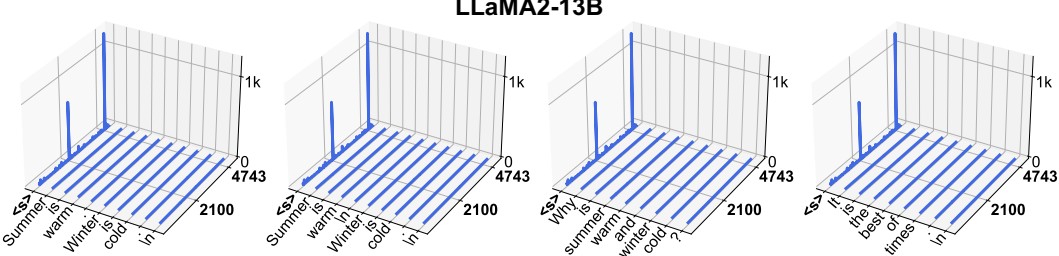

Figure 27: Massive activations in LLaMA2-13B when the input sequence is prepended with a BOS token .

### B.4 Layer-Level Analysis

In Section 2.1, we have presented the layer-level analysis results for LLaMA2-7B, LLaMA2-13B and Phi-2. In Figure 29, we provide the comprehensive results for all LLMs examined in this paper (listed in Table 8). This includes LLMs from LLaMA2, Mistral, MPT, Falcon, OPT and GPT-2 model families. For each model, we show the three largest activation magnitudes as well as the median at each layer.

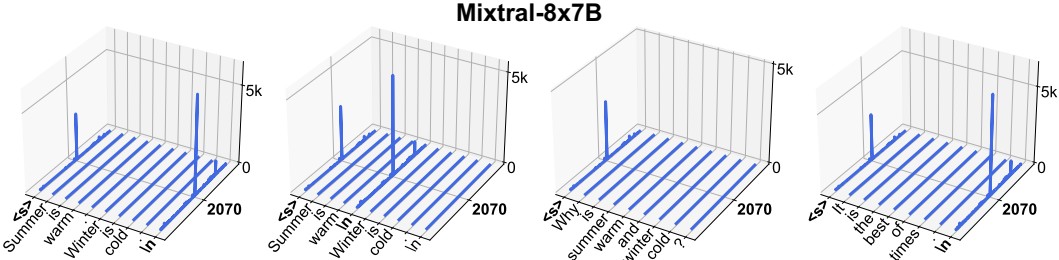

Figure 28: Massive activations in Mixtral-8x7B when the input sequence is prepended with a BOS token .

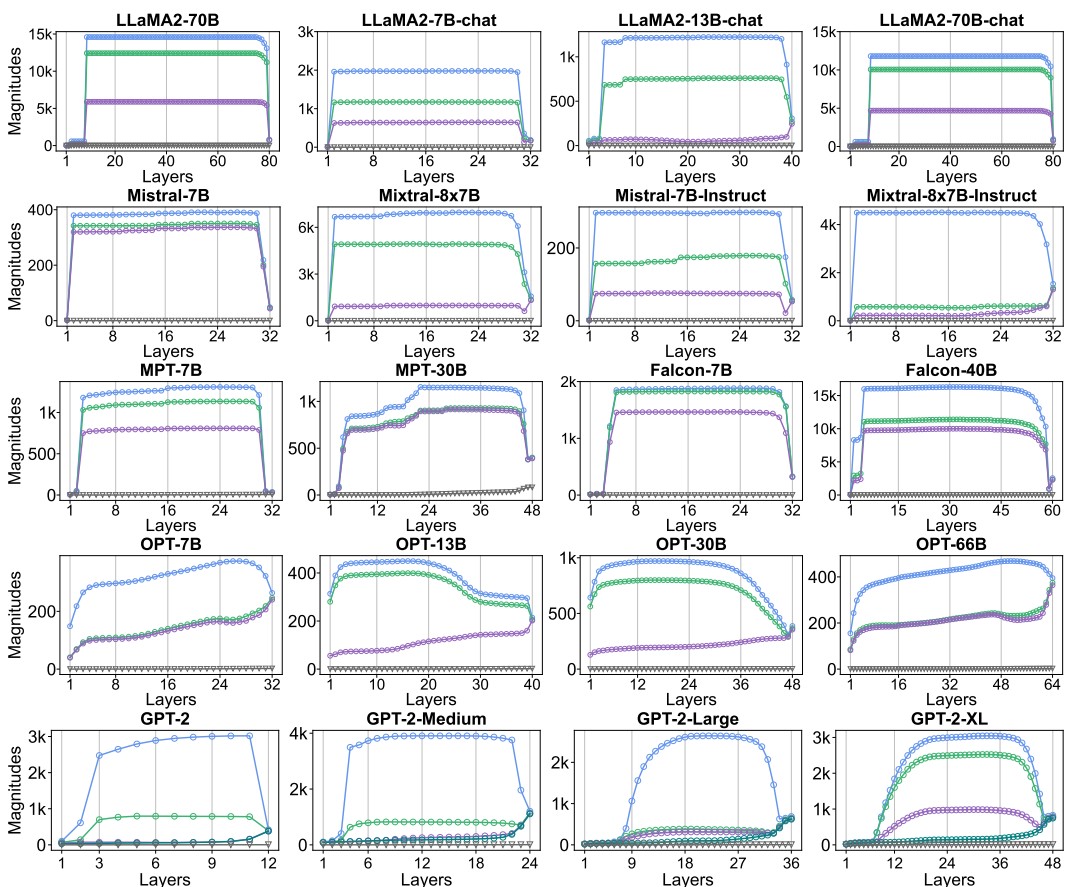

Figure 29: Layer-level analysis of LLMs. For each model, we show the three largest activation magnitudes as well as the median per layer.

We can see that the trend of massive activations we observe in Section 2.1 holds true for LLMs in general. Massive activations tend to remain constant in most of the intermediate layers. They emerge in the early layers and disappear in the last layer.

## C  Additional Results on Self-Attention

In this section, we provide additional results for the analysis on self-attention. This includes results on more LLMs (Appendix C.1), analysis of attention LayerNorm (Appendix C.2), more results on implicit attention biases (Appendix C.3) and detailed results on training GPT-2 with explicit attention biases (Appendix C.4).

### C.1  Attention Concentration on Massive Activations

In Section 4, we have demonstrated the attention concentration pattern in LLaMA2-7B, LLaMA2-13B and Phi-2. We now illustrate this phenomenon for more LLMs. Figure 30 and Figure 31 show the results for LLaMA2-70B and Mistral-7B. For these two models, massive activations are formed in the output feature of layer 9 and layer 2 respectively.

We can see that attention is predominantly focused on the sequence dimensions of massive activations. In the case of LLaMA2-70B, as depicted in Figure 30, massive activations are found in the starting word token and also token 2. These two tokens receive substantial attention logits. Additionally, we visualize the attention probability in Figure 32. The attention softmax is computed along each row, thus resulting in these special tokens being allocated a much higher attention probability.

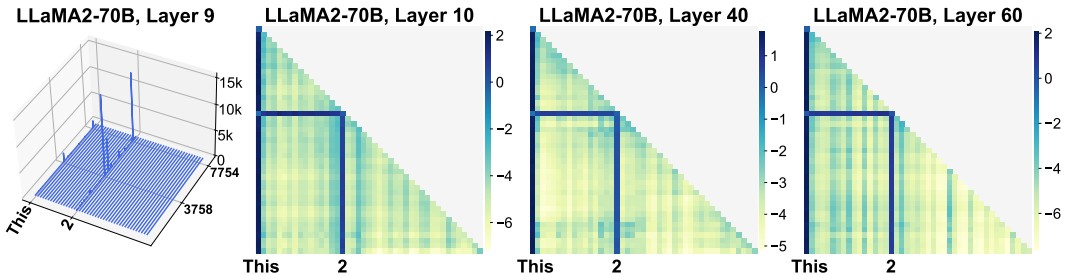

Figure 30: Average attention logits over all heads in layers 10, 40 and 60 of LLaMA2-70B. The input sequence is "*This book, including all illustrations and text, is protected under Copyright©2024 and may not be reproduced or transmitted in any form without the prior written permission of the copyright owner.*".

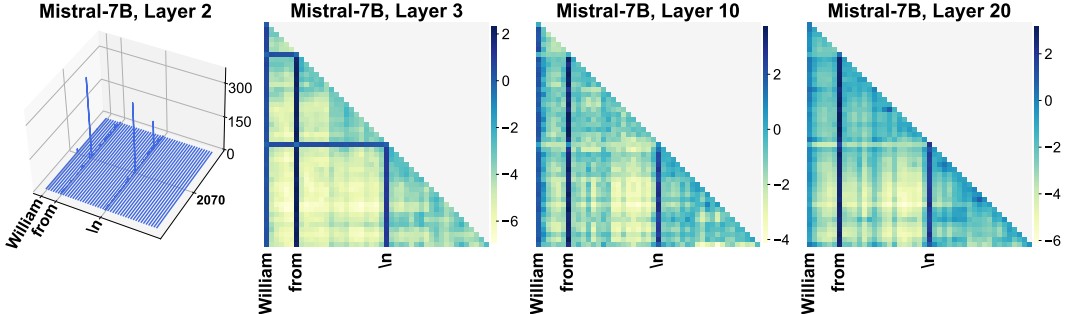

Figure 31: Average attention logits over all heads in layers 3, 10 and 20 of Mistral-7B. The input sequence is "*William Shakespeare was a famous writer from England who wrote plays and poems. He is considered one of the best writers ever.\n His works include famous plays like 'Romeo and Juliet' and 'Hamlet'.*".

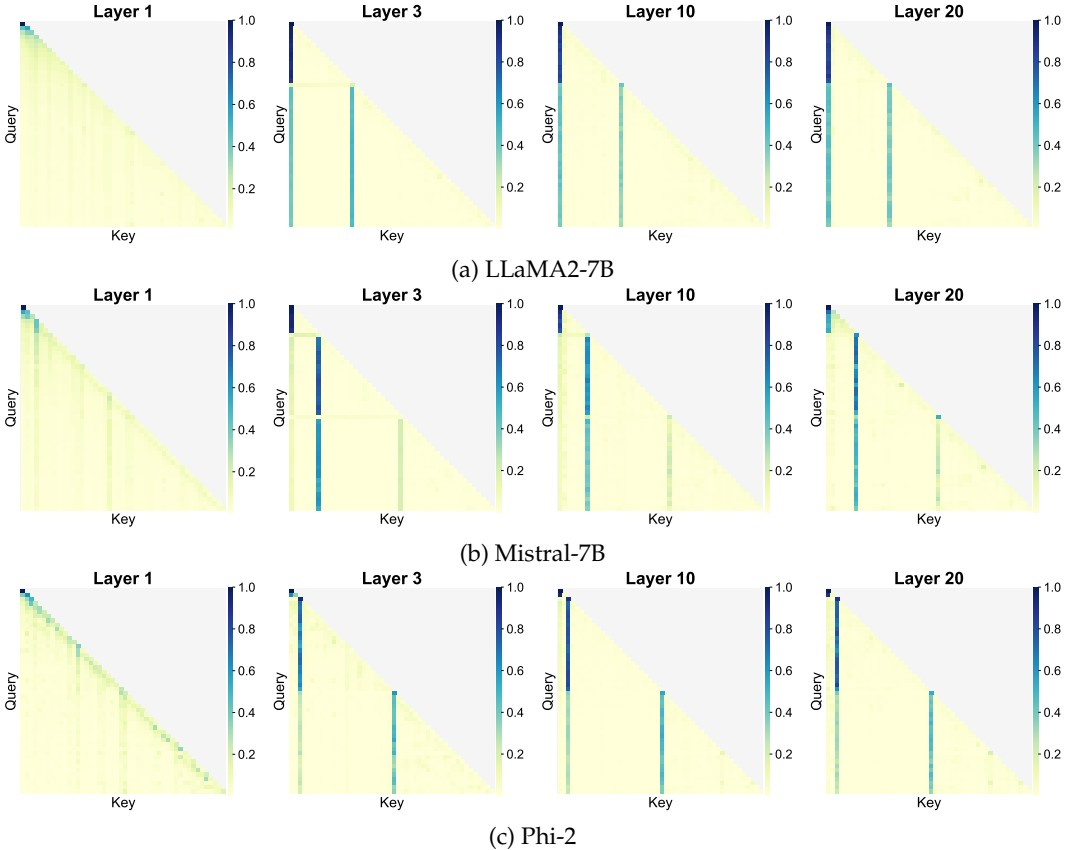

Figure 32: Average attention probability over all heads in intermediate layers of LLaMA2-7B, LLaMA2-13B and Phi-2. The input prompt is "*William Shakespeare was a famous writer from England who wrote plays and poems. He is considered one of the best writers ever.\n His works include famous plays like 'Romeo and Juliet' and 'Hamlet'.*".

## C.2 Attention LayerNorm

Our analysis in Section 4.2 indicates that tokens associated with massive activations have drastically different key and value states. In this part, we investigate how attention layer-norm plays a crucial role in this process.

**Preliminaries.** There are two specific types of layer normalization commonly used in LLMs. One is the standard layer normalization (Ba et al., 2016). Suppose we have a feature vector $x \in \mathbb{R}^d$, LayerNorm will normalize this feature to fix the mean and variance and then re-scale with element-wise affine transformation:

$$\bar{x}_i = \frac{x_i - \mu}{\sigma} * g_i + b_i, \qquad \text{where } \mu = \frac{1}{d}\sum_{i=1}^{d} x_i, \quad \sigma = \sqrt{\frac{1}{d}\sum_{i=1}^{d}(x_i - \mu)^2}. \tag{4}$$

where $g, b \in \mathbb{R}^d$ are parameters of the affine transform, also called the gain and bias.

In addition to the original LayerNorm, a variant of layer normalization has also been used in LLaMA2 and Mistral models. Specifically, Root Mean Square Normalization (RM-SNorm) (Zhang & Sennrich, 2019) normalizes the feature $x \in \mathbb{R}^d$ with the root mean square (RMS) statistic:.

$$\bar{x}_i = \frac{x_i}{\text{RMS}(\mathbf{a})} * g_i, \quad \text{where } \text{RMS}(\mathbf{x}) = \sqrt{\frac{1}{d}\sum_{i=1}^{d} x_i^2}. \tag{5}$$

where $g \in \mathbb{R}^d$ is the gain parameter.

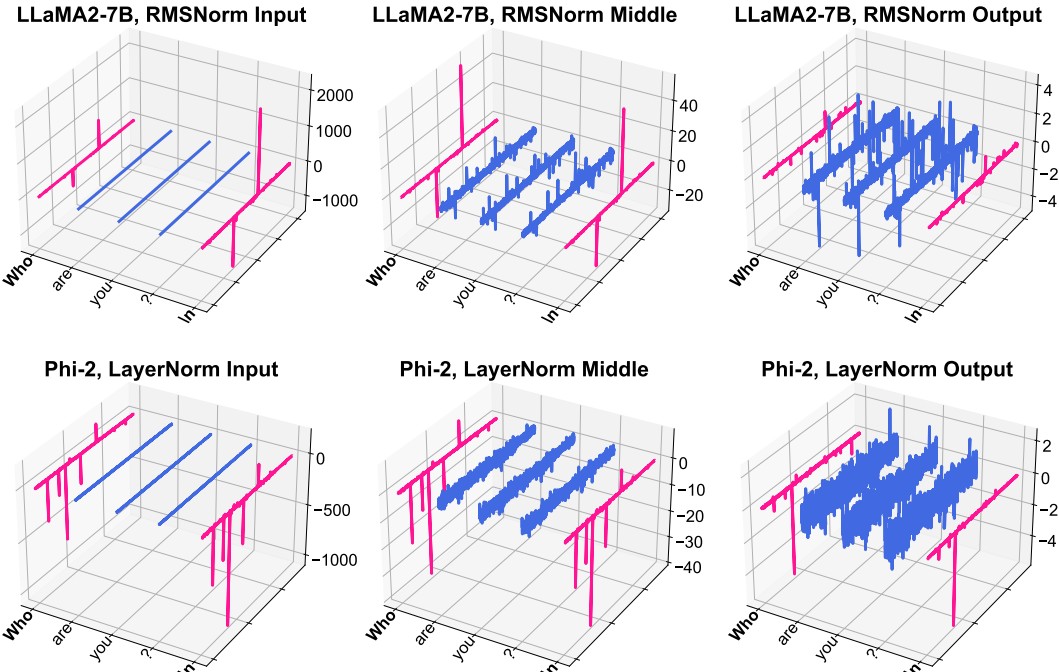

Figure 33: Activation trajectory in the attention LayerNorm of LLaMA2-7B and Phi-2, where the LayerNorm input contains massive activations. Note that LLaMA2-7B uses a variant of layer normalization: RMSNorm (Zhang & Sennrich, 2019) and Phi-2 uses the default LayerNorm (Ba et al., 2016).

For both LayerNorm and RMSNorm, when there are a few activations in $x \in \mathbb{R}^d$ that have significantly large magnitudes, the denominator in the normalization step, i.e., $\sigma$ in Equation 4 and RMS($\mathbf{x}$) in Equation 5, becomes large as a result. In fact, the denominator is almost determined by these few massive activations. The large denominator will push all normal values to zero while preserving the outlier nature of massive activations. This will effectively create a drastically different normalized feature, determined by the few massive activations. Figure 33 shows two activation trajectory in both RMSNorm and LayerNorm. We can see that how the normalization step (*middle*) preserves the outlier activations in tokens Who and \n and normalized features at these two tokens become extremely similar.

### C.3 Implicit Attention Biases

In Section 4.2, we have shown how the value updates from the tokens associated with massive activations tend to be largely identical. Here we extend those findings by examining additional input prompts and layers within the LLaMA2-7B model. We use four input prompts: "*Are you cold?\n Grab a jacket.*", "*Will it snow?\n Check the forecast.*", "*Did she call?\n I missed it.*" and ""*I am doing well. Thank you for asking.*"". We visualize the value updates in layer 3, layer 15 and layer 30 in Figure 34, Figure 35 and Figure 36 respectively. We focus on the latter half of the input sequence, following the two tokens associated with massive activations. We can see that in the same layer, the value updates $\sum_{i \in \mathcal{C}} p_i^k v_i$ display remarkable similarity across the different input sequences.

### C.4 Explicit Attention Biases

**Experimental setup.** We use the open-source reproduction of GPT-2 from the NanoGPT repository (Karpathy, 2023). We use the default recommended training setup and optimizer setting. For each of the three GPT-2 models, we train for 50,000 iterations, with a total of approximately 2B tokens. For the GPT-2 with a sink token, we follow Xiao et al. (2023b), where we prepend each training sequence with a learnable sink token [SINK]. When computing the training loss, we do not include the cross-entropy loss computed on the prepended sink token. For GPT-2 with explicit attention biases, we initialize each $\mathbf{k}'$ and $\mathbf{v}'$ with $\mathcal{N}(\mathbf{0}, 0.02\mathbf{I})$.

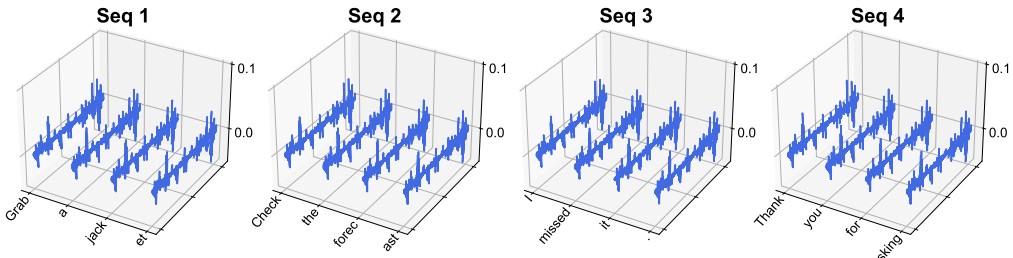

Figure 34: Value updates $\sum_{i \in \mathcal{C}} p_i^k v_i$ at layer 3 of LLaMA2-7B, with four input sequences.

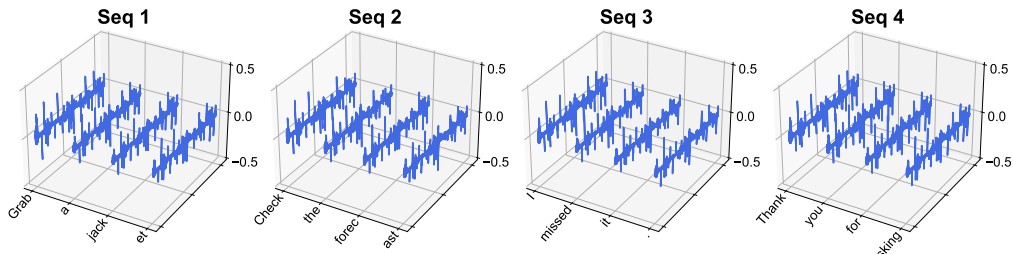

Figure 35: Value updates $\sum_{i \in \mathcal{C}} p_i^k v_i$ at layer 15 of LLaMA2-7B, with four input sequences.

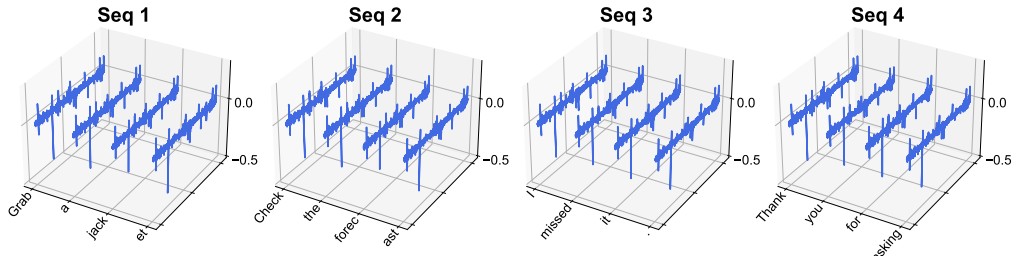

Figure 36: Value updates $\sum_{i \in \mathcal{C}} p_i^k v_i$ at layer 30 of LLaMA2-7B, with four input sequences.

**Results.** Regarding the performance of the three GPT-2 models we evaluate in Section 4.3, we find that after 50,000 training iterations, they have the same perplexity on the validation split constructed from OpenWebText2 (Gao et al., 2021): 3.04.

We also experiment with other ways of injecting biases in the self-attention computation:

1. The first one is a special case of our proposed formulation in Equation 3, where both $\mathbf{k}'$ and $\mathbf{v}'$ are zero vectors. Equation 6 shows the computation of this variant of self-attention. This is also equivalent to the previous proposed Softmax-off-by-one (Miller, 2023).

$$\text{Attention}(Q, K, V) = \text{softmax}\left(\frac{Q\begin{bmatrix} K^T & \mathbf{0} \end{bmatrix}}{\sqrt{d_k}}\right)\begin{bmatrix} V \\ \mathbf{0}^T \end{bmatrix} \tag{6}$$

2. Since Equation 3 can be viewed as inserting a sequence dimension, we also experiment with inserting one extra feature dimension. Specifically, we add learnable parameters $\mathbf{q}', \mathbf{k}' \in \mathbb{R}^T$ and concatenate them with the query and key states respectively. This variant of self-attention is as follows:

$$\text{Attention}(Q, K, V; \mathbf{q}', \mathbf{k}') = \text{softmax}\left(\frac{\begin{bmatrix} Q & \mathbf{q}' \end{bmatrix}\begin{bmatrix} K & \mathbf{k}' \end{bmatrix}^T}{\sqrt{d_k}}\right) V \tag{7}$$

3. We also experiment with a simple way to enforce constant value updates by injecting an extra value parameter $\mathbf{v}' \in \mathbb{R}^{d_k}$. This variant of self-attention is as follows:

$$\text{Attention}(Q, K, V; \mathbf{v}') = \text{softmax}\left(\frac{QK^T}{\sqrt{d_k}}\right) V + \mathbf{v}' \tag{8}$$

Figure 37 visualizes the ten largest activation magnitudes in three GPT-2 models, corresponding to the three formulations of biases in Equation 6, 7 and 8. We find that these alternatives are not able to eliminate massive activations during pretraining.

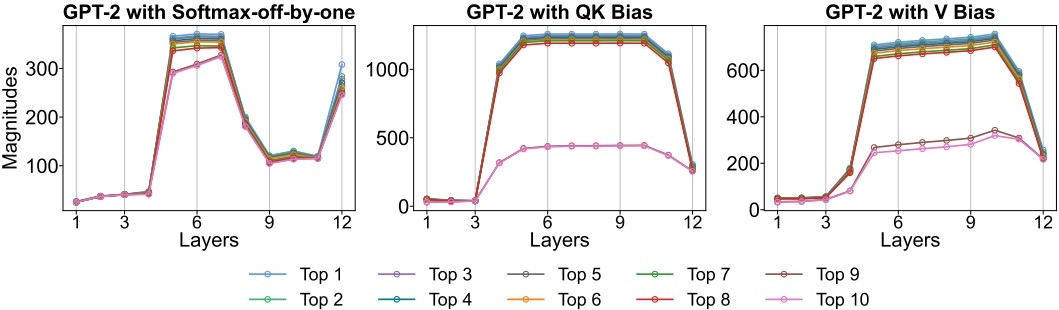

Figure 37: Ten largest activation magnitudes at each layer in three GPT-2 models.

## D   Models and Datasets

Table 8 lists the information of the LLM models used in this paper.

| Model family | Model name | Layers | Dimensions | Heads | Huggingface model id |
|---|---|---|---|---|---|
| LLaMA2 | LLaMA2-7B | 32 | 4096 | 32 | meta-llama/Llama-2-7b-hf |
| | LLaMA2-13B | 40 | 5120 | 40 | meta-llama/Llama-2-13b-hf |
| | LLaMA2-70B | 60 | 6656 | 52 | meta-llama/Llama-2-70b-hf |
| | LLaMA2-7B-Chat | 32 | 4096 | 32 | meta-llama/Llama-7b-chat-hf |
| | LLaMA2-13B-Chat | 40 | 5120 | 40 | meta-llama/Llama-2-13b-chat-hf |
| | LLaMA2-70B-Chat | 60 | 6656 | 52 | meta-llama/Llama-2-70b-chat-hf |
| Mistral | Mistral-7B | 32 | 4096 | 32 | mistralai/Mistral-7B-v0.1 |
| | Mistral-8x7B | 32 | 4096 | 32 | mistralai/Mistral-8x7B-v0.1 |
| | Mistral-7B-Instruct | 32 | 4096 | 32 | mistralai/Mistral-7B-Instruct-v0.2 |
| | Mistral-8x7B-Instruct | 32 | 4096 | 32 | mistralai/Mistral-8x7B-Instruct-v0.1 |
| Phi | Phi-2 | 32 | 2560 | 32 | microsoft/phi-2 |
| MPT | MPT-7B | 32 | 4096 | 32 | mosaicml/mpt-7b |
| | MPT-30B | 48 | 7168 | 64 | mosaicml/mpt-30b |
| Falcon | Falcon-7B | 32 | 4544 | 71 | tiiuae/falcon-7b |
| | Falcon-40B | 60 | 8192 | 128 | tiiuae/falcon-40b |
| OPT | OPT-7B | 32 | 4096 | 32 | facebook/opt-6.7b |
| | OPT-13B | 40 | 5120 | 40 | facebook/opt-13b |
| | OPT-30B | 48 | 7168 | 56 | facebook/opt-30b |
| | OPT-66B | 64 | 9216 | 72 | facebook/opt-66b |
| GPT-2 | GPT-2 | 12 | 768 | 12 | gpt2 |
| | GPT-2-Medium | 24 | 1024 | 16 | gpt2-medium |
| | GPT-2-Large | 36 | 1280 | 20 | gpt2-large |
| | GPT-2-XL | 48 | 1600 | 25 | gpt2-xl |

Table 8: Relevant information of LLM models we experimented with in this work.

We list the datasets used in this work and relevant license information:

- RedPajama (Together Computer, 2023):                    Apache License, Version 2.0

- OpenWebText2 (Gao et al., 2021):                    MIT License

- C4 (Raffel et al., 2020):                    Open Data Commons Attribution License 1.0 license

- PG-19 (Rae et al., 2019):                    Apache License, Version 2.0

- WikiText (Merity et al., 2016):                    Creative Commons BY-SA 3.0 license

- MMLU (Hendrycks et al., 2021):                    MIT License

- BoolQ (Clark et al., 2019a):             Creative Commons BY-SA 3.0 license

- PIQA (Bisk et al., 2019):             The license status is unclear

- WinoGrande (Sakaguchi et al., 2019):       Apache License, Version 2.0

- ARC easy and challenge (Clark et al., 2018):   Creative Commons BY 4.0 license

- ImageNet (Deng et al., 2009):          The license status is unclear

