# OpenReview forum: "Massive Activations in Large Language Models"
_colmweb.org/COLM/2024/Conference — COLM_

### Official Review · Reviewer_aoML · 2024-04-17

**Rating:** 8
**Confidence:** 4
**Ethics Flag:** 1

**Summary:**

This paper identifies a new phenomenon called “massive activations” in Large Language Models (LLMs), which shows that the hidden state vectors in intermediate layers at certain token locations have a few exceedingly large coordinates. The finding extends the attention sink phenomenon of Xiao et al. (2023) in the sense that these tokens are precisely the tokens that attract attention scores. The paper reports several qualitative properties such as their appearance and their difference in different LLMs. The paper performs intervention studies and further some small-scale pretraining experiments to demonstrate that massive tokens act like a fixed bias in the attention mechanism.

**Questions To Authors:**

See the “weakness” part.

**Reasons To Accept:**

Overall I think the paper reports a very intriguing and potentially important phenomenon in LLMs. The phenomenon could be of immediate interest to the theoretical understanding / mechanistic interpretability communities. The experiments are also quite comprehensive and provide a useful reference point for practitioners.

* The phenomenon extends the attention sink phenomenon and in my opinion actually gives (at least a partial) explanation of the attention sink phenomenon.

* The finding that massive tokens also appear in *delimiter tokens* (Sec 2.2) is interesting and (kind of) new even to the attention sink literature. Previously people only showed that these “information-less” tokens act as attention sink in encoder LMs or ViTs. This paper shows that delimiter tokens can also attract attention in standard autoregressive decoder LMs.

* The intuition that attention sink acts as fixed attention biases is very sensible and the paper provides some justifications.

* The study is quite comprehensive in both the models (LlaMa 7B & 13B, Mistral 7B, Phi-2) as well as the various qualitative properties.

* The presentation and in particular the visualizations are quite well done.

**Reasons To Reject:**

* The pretraining experiment (Sec 4.3) with the k’v’ variation as in Eq (3) is not fully satisfactory in showing that massive tokens act as a fixed bias.

  - Eq (3) does not really differ from the “GPT2 with sink token” baseline, as it seems mathematically almost equivalent (maybe up to some minor reparametrization, since Eq(3) did not parametrize a hidden state vector but rather one k’v’ vector for each head). So even though k’v’ method removes massive activation in other tokens as reported, I am guessing this special k’v’ token still attracts (say >90%) attention, so this method still preserves attention concentration.

  - (An interesting test would be whether k’v’ “correspond” to massive activations or their post-layernorm versions, but here it does not sound possible since you only parametrized k’v’, not the pre-KV projection versions.)

  - On the other hand, Eq (8) in appendix seems like a more ideal version of fixed attention bias, and addresses the unbalancedness in the softmax as well as the massive activation phenomenon. Unfortunately pretraining with it only works to some extent (it reduces the top activation magnitude from 2K to 600, but the ideal target is like 100-200). Again, I am wondering if the attention scores are still concentrated here (and I am guessing they are)?

  - If Eq (8) cannot be made to work to fully remove massive activation, it then creates the question of whether attention concentration is fundamental (for achieving good LM performance). Have the authors thought about other ways of removing massive activations / attention concentration?

* No immediate practical applications, but I don’t think that’s a major concern given the potential importance of the phenomenon. I also wonder if the authors have thought about alleviating training instabilities by mitigating massive activation and/or attention concentration (kind of related to the above question)?

---

> ### Author Rebuttal · Authors · 2024-05-31
>
> We thank the reviewer for the positive assessment of our paper and the constructive comments. We are happy to address your concerns.
>
> - **I am guessing this special k’v’ token still attracts (say >90%) attention, so this method still preserves attention concentration.**
>
> We confirmed that this is indeed the case for the GPT-2 model we trained with explicit attention biases. Our hypothesis is that attention concentration is a necessary pattern that is developed during pretraining. We will add discussion on this result in the revision.
>
> - **Eq (8) and attention concentration**
>
> In equation 8, there are no changes in the softmax computation except that we are adding one parametrized value embedding after value aggregation. We find that this formulation does not alter the attention concentration patterns as compared to the original GPT-2 model.
>
> - **Other ways of removing massive activations / attention concentration**
>
> Our results seem to suggest that attention concentration is a crucial pattern developed during pretraining. Thus removing this pattern might impact model performance to a large degree. In addition to explicit attention biases, there can be other ways to remove massive activations. For example, one could apply l1 regularization on the hidden states across all layers, though this extra regularization term could make pretraining trajectory much different from standard pretraining. One advantages of using explicit attention biases is that it induces minimal changes on the Transformer architecture.
>
> - **Training stabilities**
>
> We believe this is an important question and our attention bias formulation still needs to be tested at a larger model scale, e.g. 7B or larger. We will discuss this direction in the revision.
>
> If you have other questions, we are happy to answer.

---

> > ### Comment · Reviewer_aoML · 2024-06-05
> >
> > Thank you for the response! I am happy to maintain my score.

---

### Official Review · Reviewer_FDqV · 2024-05-09

**Rating:** 4
**Confidence:** 4
**Ethics Flag:** 1

**Summary:**

This paper introduces an outrageously large value sometimes appears in a hidden vector in a large language model. This paper calls such a large value "massive activation". The authors conducted several analyses on the massive activations. Then, the authors show situation which the massive activation appears in, and the relation between attention calculation and the massive activation.

**Reasons To Accept:**

The massive activations, that are introduced by this paper, are interesting phenomena. The analyses on massive activations might have an influence on future studies on LLM behaviors and training dynamics of LLMs.

**Reasons To Reject:**

This paper introduces an interesting phenomenon, the massive activation, but I think this is an ongoing study because this paper does not reveal the reason of the massive activation and the effect on LLM behaviors. For example, the relation on the stabilization of LLMs. Section 4 of this paper implies that the massive activations have a role such as a bias term. PaLM paper (https://arxiv.org/abs/2204.02311) clams that it important to remove bias terms to stabilize the pre-training of LLMs. These suggest that massive activations might have a serious influence on the stabilization of LLMs. Like this example on the stabilization, the authors should conduct more investigations to reveal why massive activations appear in.

---

> ### Author Rebuttal · Authors · 2024-05-31
>
> We thank the reviewer for the review and the constructive comments. We are happy to address your concerns.
>
> - **This paper does not reveal the reason of the massive activation and the effect on LLM behaviors.**
>
> Regarding the reason of the massive activations, in Section 4, we have showed that massive activations are fundamentally connected to the self-attention mechanism. Specifically, LLMs use these activations to allocate significant at their tokens (Section 4.1) and further form an implicit bias term in the attention output (Section 4.2). With respect to the effect on LLM behaviors, in Section 3, we have showed that massive activations play an important role on the output behaviors of LLMs.
>
> We acknowledge that our analysis in this paper is largely focused on the pretrained LLMs, where the model has already learned these massive activations during pretraining. We believe that studying the aspects of massive activations regarding pretraining would be interesting future direction, as we have showed that they can be found across LLM model families (Section 2). We will be including discussion on this direction in the updated version.
>
>
> - **bias term suggested by the PaLM paper [1]**
>
> We would like to clarify that the bias term that PaLM paper refers to is a different concept of the attention bias formulation in our paper. Specifically, PaLM paper states that “No biases were used in any of the dense kernels or layer norms. We found this to result in increased training stability for large models.”. This refers specifically to the additive bias terms in the linear layers. In the related work part of our submission, i.e., “biases in self-attention”, we have distinguished this notion of bias to our formulation of attention bias. We will be making this distinction clear in the early parts of our paper.
>
> If you have other questions, we are happy to answer.
>
> [1] PaLM: Scaling Language Modeling with Pathways. Chowdhery et al, 2022.

---

> > ### Comment · Reviewer_FDqV · 2024-06-05
> >
> > Thank the authors for their response.
> > I thought I had understood the explanation by authors. In my understanding, the authors provided analyses on the massive activations through the observation of the trained language models but I would like the authors to reveal theoretical reason why massive activations appear in. In particular, I would like the authors to provide the mathematical proof why the large language models contains the massive activations.
> >
> > However, through reading other reviews, my assigned score might be low although the paper reporting a new interesting phenomena. So, I raise my score.

---

### Official Review · Reviewer_sAPY · 2024-05-12

**Rating:** 7
**Confidence:** 3
**Ethics Flag:** 1

**Summary:**

This analysis paper studies the phenomenon of massive activations, which is defined in this paper as activations with magnitude larger than 100 and 1,000 times larger than the median magnitude of its hidden state. The paper had the following main findings:

- Massive activations widely exists in different models, largely in the middle layers, and occurs on fixed dimensions of a model and often on frequent tokens
- Setting massive activation to 0 significantly changes model behaviors.
- Massive activation causes attention to be concentrated, and act as bias terms in attention modules. Training models with a bias term in the attention module can eliminate massive activations.

In summary, I think the general presentation quality is very high, forming a very well-motivated and coherent story that's mostly easy to follow. I do have some doubts about the impact of this work, because I fail to see how such understanding can help us make improvements to the current transformer architecture. I'm hoping the authors to shed more lights on those doubts.

**Questions To Authors:**

- For "setting massive activation to a certain value" experiment in Section 3, did you try to set it to a non-zero regular magnitude value (e.g. mean of the hidden state)? I wonder if 0 has some kind mathematical specificity here that might have caused the significant change.
- For Section 4.2, I'm not quite sure if I'm following the transition between layer-norm to output decomposition -- they feel somewhat unrelated to each other. Can you explain how one affect another? Also, have you tried to run experiments with the attention module in the original transformer paper? I believe there is no layer-norm before the linear transformation.

**Reasons To Accept:**

- This paper presents a novel, interesting, and thorough analysis that advances the communities understanding of massive activation phenomenon in the transformer architecture.
- The presentation quality is very high. The clarity of the experimental setting is good. The transitions in between different parts of the analysis are well motivated. The writing is mostly easy to follow.

**Reasons To Reject:**

- Although the authors provided great insight into reasons why massive activations occur in transformers, I have some doubts on what steps the community can take on those findings. Some discussions around that theme would be very helpful -- for example, should people start to add a bias term on the attention modules? Should people be more careful with normalization in the attention module? What improvements should people expect from them? Is it better performance or just better training stability? Answering these questions can potentially help amplify the impact of the paper.
- Analysis stopped at 13B models and decoder-only models. I would be curious to see if the phenomenon generalizes to encoder-decoder models, or even larger models, e.g. 175B BLOOM models.

---

> ### Author Rebuttal · Authors · 2024-05-31
>
> We thank the reviewer for the positive assessment and constructive comments. We are happy to address your concerns.
>
> - **Practical implications**
> There could be several practical implications. First, removing massive activations would make the activations easier to quantize. Second, dedicating the learnable k’ and v’ vectors could make the attention pattern more interpretable, i.e., avoiding allocating excessive attention to input tokens. Further, we believe that studying the effect of our attention formulation on large-scale pretraining would be an important future direction.
>
> - **Analysis stopped at 13B models**
> In our paper, we have analysis on LLMs with sizes larger than 13B. The Mixtral-8x7B model in Figure 3 has 47B parameters. In the appendix, we have results on LLaMA2-70B. We find that massive activations consistently appear across model sizes.
>
> - **Encoder-decoder models**
> Following Table 1, we show the top activation magnitudes for two T5 models at one layer. We find that massive activations can be found in encoder-decoder LLMs.
> |Model| top 1 | top 2 | top 3 | top 4 | top 5 | top 10 | top 100 | top 1% | top 10% | median |
> |---|---|---|---|---|---|---|---|---|---|---|
> |T5-base| **1.3e4**|**1.1e4**|**1.1e4**|**8.6e3**|**5.2e3**|1.6e3|427|270|84|31|
> |T5-large| **1.9e4**|**8.3e3**|**7.8e3**|**7.7e3**|**7.6e3**|1.2e3|569|375|71|21|
>
> - **Setting massive activations to the mean of the hidden states**
> We conduct this intervention analysis. We find that similar to setting to zero, it leads to significant drop in performance. We observe that the mean values of the hidden states are still very small values as massive activations are extremely few in quantity.
>
> - **Layer-norm and output decomposition**
> These are two consecutive steps in the attention layer. The part on layernorm is to analyze the effect of massive activations on computing QKV states. Then attention is computed with QKV states as inputs. In the paper, the output decomposition is on the second step.
>
> - **Post-layernorm architecture**
> Since first introduced in GPT-2, pre-layernorm has become the common practice for decoder-only LLMs. We additionally trained GPT-2 with post-layernorm. We observe optimization difficulty, where the model diverges suddenly at one point during pretraining, with the training setup of default GPT-2. We are looking into this issue and will add discussion on this in the new revision.
>
> If you have other questions, we are happy to answer.

---

> > ### Comment · Reviewer_sAPY · 2024-06-05
> > **Thanks for the reponse**
> >
> > The authors have addressed all my concerns. I plan to maintain my original score.
> >
> > Please consider adding the practical implication section of the response to the paper, maybe replacing the final sentence of the paper -- it's a more meaningful way to end the paper rather than saying "we observed this, hopefully it helps" :)

---

### Official Review · Reviewer_sdg5 · 2024-05-13

**Rating:** 7
**Confidence:** 5
**Ethics Flag:** 1

**Summary:**

This paper identifies the phenomenon of massive activations in large language models (LLMs), where very few activations exhibit significantly larger values than others. It extensively studies the layers where massive activations reside, their importance to LLMs' capabilities, and their role as bias in the self-attention mechanism. Finally, the analysis is further extended to ViTs.

**Reasons To Accept:**

1. The identified existence of massive activations is interesting. Although related work like FLAP [1] has provided similar observations, a comprehensive study of this phenomenon could provide useful insights for the community.

[1] "Fluctuation-based Adaptive Structured Pruning for Large Language Models", Y. An et al., AAAI'24.

2. The performed analysis regarding the importance and role of massive activations is technically sound and extensive.

3. The writing is logically clear and coherent.

**Reasons To Reject:**

1. Some of the observations lack deeper analysis. For example, massive activations seem to serve as an important bias in the attention mechanism, while they mainly focus on semantically unimportant tokens like delimiters. The authors are expected to provide a clearer explanation regarding this gap.

2. It is not clear whether massive activations are desirable or whether they will disappear with larger-scale pretraining. The authors are expected to evaluate larger LLMs pretrained on larger corpora to determine whether the percentage of massive activations will increase with the model scale and data scale.

---

> ### Author Rebuttal · Authors · 2024-05-31
>
> We thank the reviewer for the positive assessment of our paper and the constructive comments. We are happy to address your concerns.
>
> - **Massive activations seem to serve as an important bias in the attention mechanism, while they mainly focus on semantically unimportant tokens like delimiters. The authors are expected to provide a clearer explanation regarding this gap**
>
>
> In the last part of section 3, we have discussed why massive activations appear in semantically unimportant tokens. Suppose the LLM model wants to store these important biases somewhere, these semantically unimportant tokens would be better than other tokens. This is because repurposing the hidden states of these tokens to store such biases would incur less loss of input information.
>
> - **The authors are expected to evaluate larger LLMs pretrained on larger corpora to determine whether the percentage of massive activations will increase with the model scale and data scale.**
>
> Regarding “model scale”, in the appendix, we have evaluated the largest LLaMA2-70B model (see Figure 17 in Appendix B.1). We find that compared to the smaller LLaMA2-7B and LLaMA2-13B model, the magnitudes of massive activations are drastically larger in LLaMA2-70B, e.g., 15000. We also have results on the Mistral-7B model (see Figure 19 in Appendix B.1). Compared to the larger Mistral-8x7B model (Figure 3), the scale of massive activations also increases with model sizes.
>
> To answer the question regarding the relationship between massive activations and “data scale”, we have studied how massive activaitons changes during the pretraining phase. Specifically, we attempt to address the question: does more pretraining lead to changes of massive activations? We observe that this is not the case. We conduct our analysis on GPT-2. We observe that massive activaitons grow rapidly in the initial part of the pretraining phase and stay at a constant level after the initial growing phase.
>
> In the updated version of the paper, we will add these results and discussion on the related work [1].
>
>
> If you have other questions, we are happy to answer.
>
> [1] Fluctuation-based Adaptive Structured Pruning for Large Language Models, An et al, AAAI’24.

---

> > ### Comment · Reviewer_sdg5 · 2024-06-04
> > **Response to author feedback**
> >
> > Thank the authors for their response, which properly addresses my concerns. I will keep my original rating.

---

### Decision · Program_Chairs · 2024-07-10

**Decision:**

Accept

**Comment:**

The reviewers agree that this work provides an interesting observation, has thorough experiments around it, and is well presented. Some reviewers would like more discussion of practical implications of the presence of massive activations and FDqV feels that the paper does not provide sufficient evidence, especially theoretical analysis, to understand the fundamental reasons behind these patterns. I am inclined to agree that a theoretical analysis or controlled synthetic experiments could help us understand the phenomenon better, but I view that as an opportunity for followup work.